# Diet-wide analyses for risk of colorectal cancer: prospective study of 12,251 incident cases among 542,778 women in the UK

Keren Papier [1] ✉, Kathryn E. Bradbury [2,10], Angela Balkwill[1], Isobel Barnes[1], Karl Smith-Byrne [1], Marc J. Gunter[3,4], Sonja I. Berndt[5], Loic Le Marchand[6], Anna H. Wu[7], Ulrike Peters [8], Valerie Beral[1,9], Timothy J. Key [1] & Gillian K. Reeves[1]

Uncertainty remains regarding the role of diet in colorectal cancer development. We examined associations of 97 dietary factors with colorectal cancer risk in 542,778 Million Women Study participants (12,251 incident cases over 16.6 years), and conducted a targeted genetic analysis in the ColoRectal Transdisciplinary Study, Colon Cancer Family Registry, and Genetics and Epidemiology of Colorectal Cancer Consortium (GECCO). Alcohol (relative risk per 20 g/day=1.15, 95% confidence interval 1.09-1.20) and calcium (per 300 mg/day=0.83, 0.77–0.89) intakes had the strongest associations, followed by six dairy-related factors associated with calcium. We showed a positive association with red and processed meat intake and weaker inverse associations with breakfast cereal, fruit, wholegrains, carbohydrates, fibre, total sugars, folate, and vitamin C. Genetically predicted milk consumption was inversely associated with risk of colorectal, colon, and rectal cancers. We conclude that dairy products help protect against colorectal cancer, and that this is driven largely or wholly by calcium.

Colorectal cancer is the third most common cancer in the world, with an estimated 1,926,425 incident cases in 2022[1]. The incidence rates vary markedly, with higher rates in high income countries including most European countries, North America, Australia, New Zealand and Japan, and lower rates in low income countries including much of Africa and south Asia[1], although the rates in lower incidence areas appear to be increasing[2]. In addition, colorectal cancer rates in migrants have been shown to change within as little as just over a decade towards those of their adopted country[3], indicating that lifestyle and environmental factors are involved in the aetiology of this cancer.

The International Agency for Research on Cancer (IARC) has classified alcoholic beverages and processed meat as carcinogenic to humans (Group 1) and red meat as probably carcinogenic (Group 2 A), with the evidence for this classification being based partly (alcohol), or largely (red and processed meat), on the findings for colorectal cancer. The World Cancer Research Fund (WCRF)/American Institute for Cancer Research (AICR)'s third expert report similarly concluded that there is convincing evidence that higher intakes of alcohol and processed meat increase the risk of colorectal cancer; they also concluded that higher consumption of dairy products, dairy milk, calcium,

[1]Cancer Epidemiology Unit, Nuffield Department of Population Health, University of Oxford, Oxford, UK. [2]Department of Epidemiology and Biostatistics, School of Population Health, Faculty of Medical and Health Sciences, The University of Auckland, Auckland, New Zealand. [3]Cancer Epidemiology and Prevention Research Unit, Department of Epidemiology and Biostatistics, School of Public Health, Imperial College London, London, UK. [4]Nutrition and Metabolism Branch, International Agency for Research on Cancer, Lyon, France. [5]Division of Cancer Epidemiology and Genetics, National Cancer Institute, National Institutes of Health, Bethesda, MD, USA. [6]University of Hawaii Cancer Center, Honolulu, HI, USA. [7]University of Southern California, Department of Population and Public Health Sciences, Los Angeles, CA, USA. [8]Division of Public Health Sciences, Fred Hutchinson Cancer Center, Seattle, WA, USA. [9]Deceased: Valerie Beral. [10]These authors contributed equally: Keren Papier, Kathryn E. Bradbury. ✉e-mail: Keren.Papier@ndph.ox.ac.uk

calcium supplements, wholegrains, and foods containing dietary fibre "probably" reduce the risk of colorectal cancer, while higher intake of red meat "probably" increases risk. The evidence for other foods, nutrients and beverages is inconclusive[4–7].

The lack of consensus regarding the relationships between dietary factors other than alcohol and processed meat and colorectal cancer risk may be due, at least in part, to the relatively few studies publishing comprehensive results on all food types[4], dietary measurement error[7,8], and/or small sample sizes[4]. In order to address some of these limitations, we report here on a systematic analysis of 97 dietary factors and subsequent colorectal cancer risk using a diet-wide association study[5,9] based on data from a large prospective study of 542,778 UK women who completed a detailed dietary questionnaire, of whom 7% also completed at least one subsequent 24-hour online dietary assessment. We also present complementary findings from a Mendelian randomisation analysis of milk consumption, using data from the ColoRectal Transdisciplinary Study, the Colon Cancer Family Registry, and the Genetics and Epidemiology of Colorectal Cancer consortium (GECCO).

## Results

These 542,778 women had a mean (SD) of 16.6 (4.8) years of follow-up, during which 12,251 women were diagnosed with incident colorectal cancer. Table 1 shows participants' characteristics overall and according to whether they developed incident colorectal cancer during follow-up. Women who developed colorectal cancer were older, taller, had more family history of bowel cancer, and had more adverse health behaviours compared with participants overall. Figure 1 and Supplementary Data 1 show the RRs for colorectal cancer in relation to intakes of the 97 dietary factors, of which 17 were associated with risk of colorectal cancer (FDR corrected $p$-value < 0.009). Of these 17 dietary factors, alcohol and calcium intakes had the strongest associations (based on lowest $p$ value) with colorectal cancer risk; with a positive association for alcohol (relative risk [RR] per 20 g/day = 1.15, 95% confidence interval [CI] 1.09–1.20, $p$ < 0.0000001) and an inverse association for calcium (RR per 300 mg/day = 0.83, 95% CI 0.77–0.89, $p$ < 0.000001). Dairy milk, yogurt, riboflavin, magnesium, phosphorus, and potassium intakes were inversely associated with colorectal cancer risk, as were intakes of breakfast cereal, fruit, wholegrains, carbohydrates, fibre, total sugars, folate, and vitamin C. Red and processed meat intake was positively associated with risk of colorectal cancer (per 30 g/day = 1.08, 1.03–1.12, $p$ < 0.01). For all of these 17 dietary factors, the categorical RRs and 95% CIs were broadly consistent with their respective log-linear dose response relationships (Figs. 2 and 3).

The pairwise correlations for the 17 FDR-significant dietary factors are displayed in Table 2. Dairy-related foods and nutrients had the strongest pairwise correlations (calcium, phosphorus, riboflavin, dairy milk, magnesium, potassium). We also observed strong and moderately-strong (r > 0.5) pairwise correlations between fibre-related foods and nutrients (including carbohydrates, total sugars, magnesium, fibre, folate, wholegrains, vitamin C, and fruit). Dairy-related and fibre-related foods and nutrients also had moderate (r > 0.35) pairwise correlations between them. In contrast, alcohol, and red and processed meat were generally only very weakly correlated with the other dietary factors.

Sequential adjustment for different types of potential lifestyle confounders across the models did not materially change the log relative risks for those dietary factors which showed the most statistically significant associations (based on lowest $p$ values) with risk of colorectal cancer (alcohol, calcium, dairy milk). Conversely, progressive adjustment led to attenuation of the magnitude of the associations with other FDR-significant dietary factors (including fruit, wholegrains, breakfast cereal), suggesting that the associations with risk of colorectal cancer for these latter foods may have been at least partly due to residual confounding with lifestyle factors (Supplementary Data 2).

**Table 1 | Characteristics of 542,778 women at baseline, and details of follow-up**

| Characteristics, mean (SD) or num (%) | All women N = 542,778 | Cases N = 12,251 | Non-cases N = 530,527 |
|---|---|---|---|
| Socio-demographic | | | |
| Age at dietary assessment, y, mean (SD) | 59.7 (4.9) | 61.1 (5.1) | 59.7 (4.9) |
| Area-based deprivation quintiles, low to high n (%) | | | |
| 1 | 126237 (23.4%) | 2883 (22.6%) | 123354 (23.5%) |
| 2 | 120532 (22.4%) | 2860 (22.5%) | 117672 (22.4%) |
| 3 | 112420 (20.9%) | 2710 (21.3%) | 109710 (20.9%) |
| 4 | 101076 (18.8%) | 2457 (19.3%) | 98619 (18.7%) |
| 5 | 78466 (14.6%) | 1824 (14.3%) | 76642 (14.6%) |
| Education attainment, n (%) | | | |
| None | 179561 (33.6%) | 4279 (34.0%) | 175282 (33.6%) |
| Technical | 91852 (17.2%) | 2103 (16.7%) | 89749 (17.2%) |
| Secondary | 170364 (31.9%) | 4059 (32.2%) | 166305 (31.9%) |
| Tertiary | 92510 (17.3%) | 2149 (17.1%) | 90361 (17.3%) |
| Lifestyle | | | |
| Strenuous exercise, n (%) | | | |
| none | 221853 (41.7%) | 5492 (43.8%) | 216361 (41.7%) |
| ≤1 per week | 184820 (34.8%) | 4215 (33.6%) | 180605 (34.8%) |
| >1 per week | 124830 (23.5%) | 2832 (22.6%) | 121998 (23.5%) |
| Smoking, n (%) | | | |
| Never | 297177 (55.5%) | 6804 (54.0%) | 290373 (55.6%) |
| Past | 176474 (33.0%) | 4325 (34.3%) | 172149 (32.9%) |
| Current smoker | 61559 (11.5%) | 1474 (11.7%) | 60085 (11.5%) |
| Alcohol, drinks per week, n (%) | | | |
| 0 | 186872 (34.4%) | 4470 (34.9%) | 182402 (34.4%) |
| 1–5 | 174278 (32.1%) | 3903 (30.5%) | 170375 (32.1%) |
| 6–10 | 110878 (20.4%) | 2605 (20.3%) | 108273 (20.4%) |
| 11 or more | 70750 (13.0%) | 1832 (14.3%) | 68918 (13.0%) |
| Energy intake, kJ per day, mean (SD) | 8194 (2327) | 6995.0 (1732.4) | 6980.6 (1753.4) |
| Health | | | |
| BMI, kg/m², mean (SD) | 25.9 (4.4) | 25.9 (4.4) | 25.9 (4.4) |
| Height, n (%) | | | |
| <160 cm | 160679 (29.9%) | 3495 (27.6%) | 157184 (30.0%) |
| 160–164 cm | 162834 (30.3%) | 3725 (29.4%) | 159109 (30.3%) |
| ≥165 cm | 213799 (39.8%) | 5445 (43.0%) | 208354 (39.7%) |
| HTM use, n (%) | | | |
| ever | 286840 (53.6%) | 6258 (49.6%) | 280582 (53.7%) |
| never | 248444 (46.4%) | 6353 (50.4%) | 242091 (46.3%) |
| Family history of bowel cancer, n (%) | | | |
| none | 495252 (91.2%) | 11389 (88.9%) | 483863 (91.3%) |
| yes | 47526 (8.8%) | 1421 (11.1%) | 46105 (8.7%) |
| **Follow-up for colorectal cancer** | | | |
| Person-years of follow-up, mean (SD) | 16.6 (4.8) | 10.6 (5.4) | 16.8 (4.7) |

[1]No education (left at or before compulsory school leaving age), Technical (non-university qualifications e.g. nursing, teaching), Secondary (O levels or A levels), Tertiary (college or university). *HTM* Hormonal therapy for menopause.

Table 3 presents associations of the 17 FDR-significant dietary factors with risk of colorectal cancer, further adjusted for calcium, dairy milk, fruit, and wholegrains. After adjustment for calcium, the inverse associations for dairy milk, phosphorus, riboflavin, magnesium, potassium, yogurt, folate, breakfast cereal and total sugars were no longer evident. Adjustment for dairy milk attenuated the

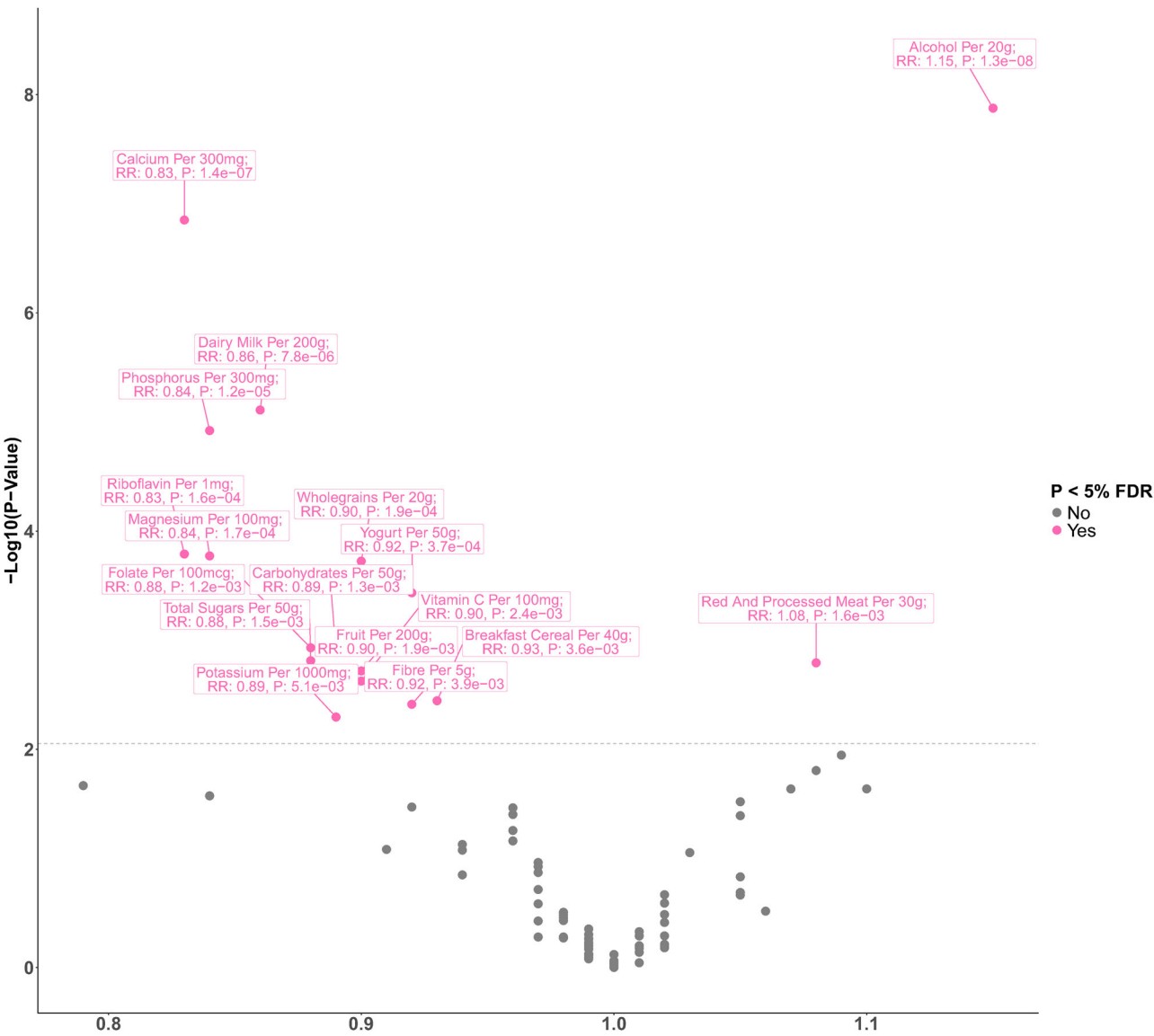

**Fig. 1 | Volcano plot showing results from diet-wide study method evaluating associations between 97 dietary risk factors and colorectal cancer risk.** The Y axis shows *p* values (two-sided) for the associations between each of the 97 dietary factors and colorectal cancer incidence calculated separately using Cox proportional hazards regression models stratified by year of birth, date of completion of the dietary survey (which is the baseline for this study), and region of residence (10 geographical regions: 9 in England and 1 in Scotland), and adjusted for area-based deprivation (fifths, based on the Townsend deprivation score, unknown), highest educational qualification (none, technical, secondary, tertiary, unknown), body mass index (< 20, 20-22.49, 22.5-24.9, 25.0-27.49, 27.5-29.9, 30-32.49, 32.5–34.9, 35+ kg/m², unknown), height (<160, 160-164.9, ≥165 cm, unknown), strenuous exercise (none, ≤ once per week, > once per week, unknown), dietary energy intake (except for the analysis of energy and risk; fifths, unknown), alcohol (none, 1–5, 6–10, ≥11 drinks per week, unknown), smoking (never, past, current 1–4, current 5–9, current <10, current 10–14, current 15–19, current 20–24, current 25–29, current ≥ 30 cigarettes per day, unknown), current use of hormonal therapy for menopause (no, yes, unknown), and family history of bowel cancer (no, yes). The X axis shows relative risks (see Supplementary Data 1 for increments). Dietary factors associated with risk of colorectal cancer with a False Discovery Rate (FDR) *p* value < 0.05 are shaded in pink and those with a *p* > 0.05 are shaded in grey. For each of the 62 quantitatively measured dietary factors, we created a continuous variable using the re-measured mean intakes for each baseline category. Log-linear trends in risk across categories of baseline intakes were then calculated using the listed increments.

associations for riboflavin, breakfast cereal, and potassium to a lesser extent than did adjustment for calcium and did not completely explain the association of calcium intake with risk, which remained significant. Adjustment for fruit intake also led to attenuation of the associations for phosphorus, riboflavin, magnesium, potassium, folate, carbohydrates, total sugars, vitamin C, breakfast cereal, and fibre, and adjustment for wholegrains attenuated the associations for magnesium, breakfast cereal and fibre, to the extent that none of these associations remained significant after adjustment. Adjusting for calcium, dairy milk, fruit, or wholegrains minimally affected the associations for wholegrains, alcohol and red and processed meat (Table 3).

Given the high correlation between calcium and dairy milk, we investigated the association of each, independently of the other, using the residuals method. Calcium intake was independently associated with risk of colorectal cancer whereas dairy milk intake was not; LRT = 6.39 (*p* = 0.01) between the models including dairy milk with or without addition of the estimated residuals for calcium intake, and LRT = 0.18 (*p* = 0.67) between the models including calcium with or without addition of the estimated residuals for dairy milk intake. We also investigated the associations of dietary calcium with risk of colorectal cancer by source of dietary calcium and found no evidence of heterogeneity by source (phet=0.21) Table 4.

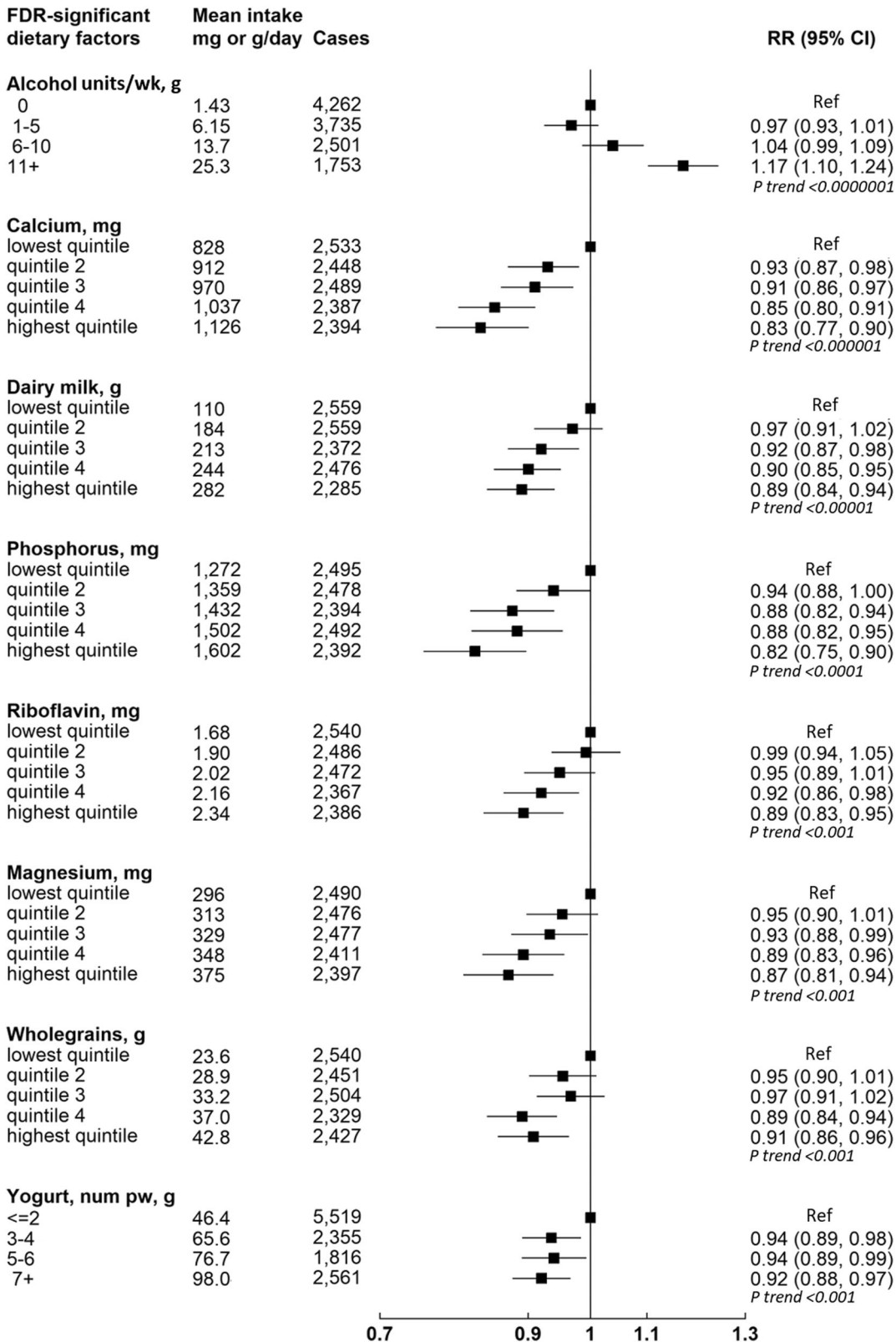

**Fig. 2 | Associations of the top eight FDR-significant dietary factors (*p* < 0.001) and colorectal cancer by intake categories.** Mean daily intakes taken from the mean of the 24-hour dietary assessments. Wholegrain intake represents actual grams of wholegrains. Associations between each of the foods or nutrients and colorectal cancer incidence calculated separately using Cox proportional hazards regression models stratified by year of birth, date of completion of the dietary survey (which is the baseline for this study), and region of residence (10 geographical regions: 9 in England and 1 in Scotland), and adjusted for area-based deprivation (fifths, based on the Townsend deprivation score, unknown), highest educational qualification (none, technical, secondary, tertiary, unknown), body mass index (< 20, 20–22.49, 22.5–24.9, 25.0–27.49, 27.5–29.9, 30–32.49, 32.5–34.9, 35+ kg/m², unknown), height (< 160, 160–164.9, ≥ 165 cm, unknown), strenuous exercise (none, ≤ once per week, > once per week, unknown), dietary energy intake (except for the analysis of energy and risk; fifths, unknown), alcohol (none, 1–5, 6–10, ≥ 11 drinks per week, unknown), smoking (never, past, current 1–4, current 5–9, current <10, current 10–14, current 15–19, current 20–24, current 25–29, current ≥30 cigarettes per day, unknown), current use of hormonal therapy for menopause (no, yes, unknown), and family history of bowel cancer (no, yes).

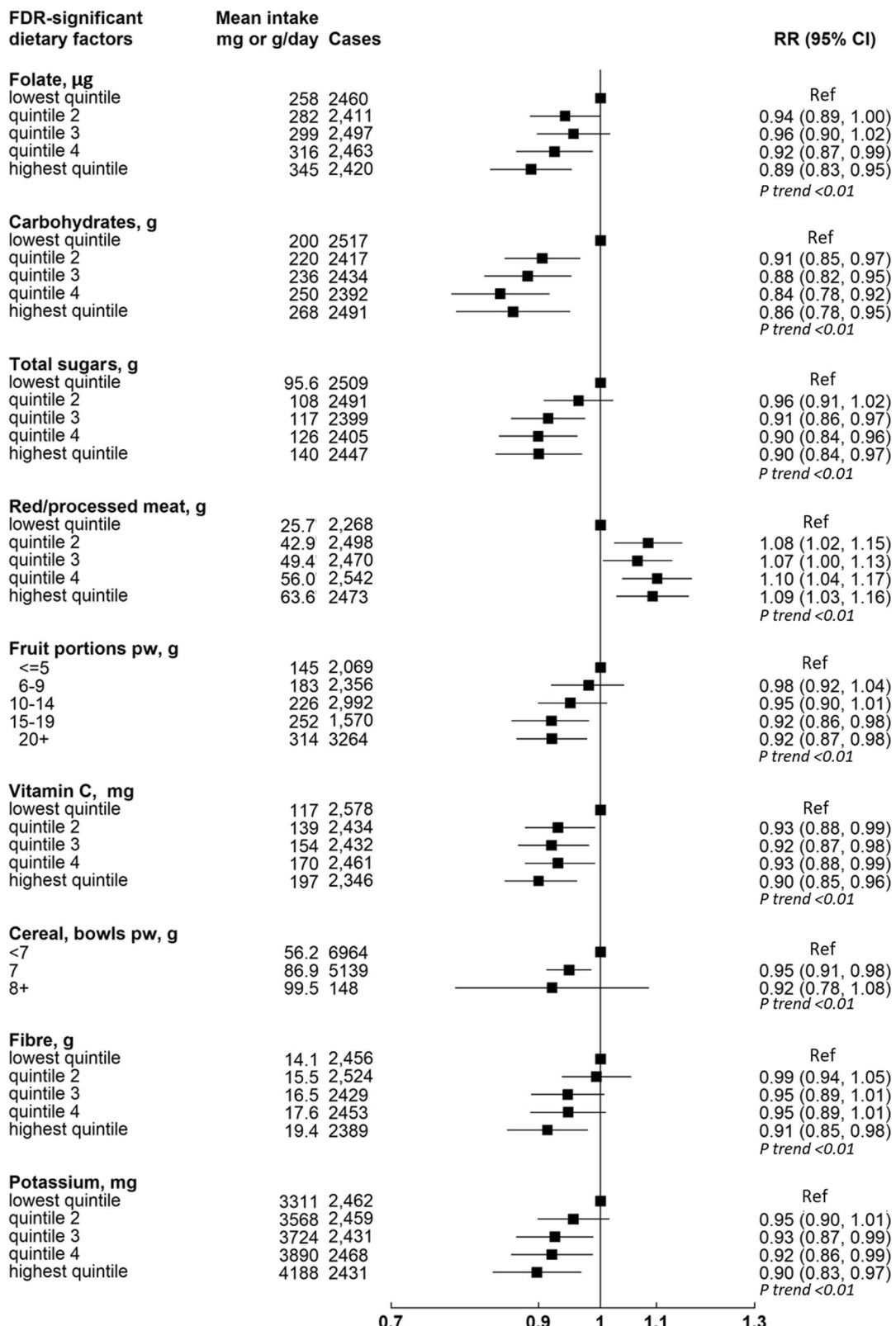

**Fig. 3 | Associations of the nine less FDR-significant dietary factors and colorectal cancer ($p < 0.01$) by intake categories.** Mean daily intakes taken from the mean of the 24 h dietary assessments. Associations between each of the foods or nutrients and colorectal cancer incidence calculated separately using Cox proportional hazards regression models stratified by year of birth, date of completion of the dietary survey (which is the baseline for this study), and region of residence (10 geographical regions: 9 in England and 1 in Scotland), and adjusted for area-based deprivation (fifths, based on the Townsend deprivation score, unknown), highest educational qualification (none, technical, secondary, tertiary, unknown), body mass index ($< 20$, 20–22.49, 22.5–24.9, 25.0–27.49, 27.5–29.9, 30–32.49, 32.5–34.9, 35+ kg/m$^2$, unknown), height ($< 160$, 160–164.9, $\geq 165$ cm, unknown), strenuous exercise (none, $\leq$ once per week, $>$ once per week, unknown), dietary energy intake (except for the analysis of energy and risk; fifths, unknown), alcohol (none, 1–5, 6–10, $\geq$ 11 drinks per week, unknown), smoking (never, past, current 1–4, current 5–9, current <10, current 10–14, current 15–19, current 20–24, current 25–29, current $\geq$30 cigarettes per day, unknown), current use of hormonal therapy for menopause (no, yes, unknown), and family history of bowel cancer (no, yes).

**Table 2 | Pairwise correlations among FDR-significant dietary factors (by order of p value for association)**

| | Alcohol | Calcium | Dairy milk | Phosphorus | Riboflavin | Magnesium | Wholegrains | Yogurt | Folate | Carbohydrates | Total sugars | Red and processed meat | Fruit | Vitamin C | Breakfast cereal | Fibre | Potassium |
|---|---|---|---|---|---|---|---|---|---|---|---|---|---|---|---|---|---|
| Alcohol | 1 | | | | | | | | | | | | | | | | |
| Calcium | -0.07 | 1 | | | | | | | | | | | | | | | |
| Dairy milk | -0.14 | 0.74 | 1 | | | | | | | | | | | | | | |
| Phosphorus | 0 | 0.89 | 0.60 | 1 | | | | | | | | | | | | | |
| Riboflavin | -0.05 | 0.82 | 0.68 | 0.86 | 1 | | | | | | | | | | | | |
| Magnesium | 0.09 | 0.74 | 0.47 | 0.89 | 0.73 | 1 | | | | | | | | | | | |
| Wholegrains | -0.02 | 0.27 | 0.13 | 0.46 | 0.33 | 0.57 | 1 | | | | | | | | | | |
| Yogurt | -0.08 | 0.40 | 0.06 | 0.35 | 0.36 | 0.32 | 0.16 | 1 | | | | | | | | | |
| Folate | 0.02 | 0.57 | 0.32 | 0.71 | 0.74 | 0.69 | 0.34 | 0.23 | 1 | | | | | | | | |
| Carbohydrates | -0.14 | 0.64 | 0.36 | 0.71 | 0.58 | 0.67 | 0.35 | 0.29 | 0.60 | 1 | | | | | | | |
| Total sugars | -0.13 | 0.62 | 0.36 | 0.61 | 0.54 | 0.58 | 0.21 | 0.37 | 0.49 | 0.87 | 1 | | | | | | |
| Red and processed meat | 0.09 | 0.10 | 0.05 | 0.32 | 0.20 | 0.17 | -0.04 | 0.06 | 0.19 | 0.13 | 0.06 | 1 | | | | | |
| Fruit | -0.03 | 0.21 | -0.01 | 0.28 | 0.23 | 0.43 | 0.25 | 0.21 | 0.40 | 0.33 | 0.44 | -0.06 | 1 | | | | |
| Vitamin C | 0.04 | 0.26 | 0 | 0.33 | 0.25 | 0.47 | 0.22 | 0.18 | 0.59 | 0.39 | 0.49 | 0.02 | 0.64 | 1 | | | |
| Breakfast cereal | -0.13 | 0.26 | 0.32 | 0.33 | 0.41 | 0.32 | 0.43 | 0.15 | 0.32 | 0.30 | 0.22 | -0.03 | 0.15 | 0.11 | 1 | | |
| Fibre | -0.02 | 0.40 | 0.12 | 0.62 | 0.46 | 0.75 | 0.66 | 0.20 | 0.73 | 0.60 | 0.49 | 0.07 | 0.66 | 0.63 | 0.37 | 1 | |
| Potassium | 0.05 | 0.71 | 0.49 | 0.84 | 0.71 | 0.93 | 0.34 | 0.32 | 0.72 | 0.67 | 0.63 | 0.26 | 0.50 | 0.56 | 0.23 | 0.69 | 1 |

### Sensitivity analyses

In sensitivity analyses restricted to women who self-reported excellent or good health at baseline, and to risk of colorectal cancer in the period 5 or more years after baseline, the findings were broadly similar (Supplementary Data 3). We also found no significant heterogeneity in the observed associations for the FDR-significant dietary factors by cancer sub-site, except for alcohol, which appeared to be less detrimental in the proximal colon and most harmful in the rectum (Supplementary Data 4; p for heterogeneity by subsite=0.02). In analyses stratified by smoking status, BMI category, area-based deprivation and alcohol intake, we observed stronger associations for dairy milk (p for heterogeneity =0.005) and riboflavin (p for heterogeneity =0.006) in never smokers and a stronger association for wholegrains (p for heterogeneity =0.04) in those with a lower BMI (Supplementary Data 5).

### MR using lactase polymorphism SNP

We observed an inverse association of genetically predicted milk consumption and risk of colorectal cancer that was larger than the inverse association with reported dairy milk intake and colorectal cancer: RR per 200 g/day=0.60, 0.46–0.74; colon cancer: RR per 200 g/day=0.60, 0.43–0.77; and rectal cancer: RR per 200 g/day=0.49, 0.31–0.67.

## Discussion

In this large prospective study of diet and colorectal cancer, we found a marked positive association for alcohol, and a strong inverse association for calcium. Inverse associations were also observed with other dairy-related factors including dairy milk, yogurt, riboflavin, magnesium, phosphorus, and potassium which, on further analysis, appeared to be primarily due to the association of these dietary factors with calcium. Further evidence for a potentially causal role for calcium in colorectal cancer incidence was provided by an accompanying analysis of genetically-predicted milk intake, which is likely to also reflect calcium intake. We also found a positive association for red and processed meat intake that was minimally affected by confounding by diet and lifestyle factors. In addition, we observed inverse associations with risk for breakfast cereal, fruit, wholegrains, carbohydrates, fibre, total sugars, folate, and vitamin C, but these inverse associations may have been influenced by residual confounding by lifestyle and/or other dietary factors.

Our study recapitulates the well-established positive association between alcohol consumption and risk of colorectal cancer[4] and is in line with the 2018 WCRF dose-response meta-analysis which found a seven percent higher risk of colorectal cancer per 10 grams of alcohol per day (equivalent to 14% per 20 grams of alcohol per day)[4] which is of nearly identical magnitude to the 15% higher risk we observed per 20 grams per day. Previous MR studies in adults of Asian and European ancestry also support a causal association of alcohol intake with colorectal cancer risk[10-12]. Suggested mechanisms by which alcohol could increase the risk of colorectal cancer include the production of acetaldehyde, found to be mutagenic in high concentrations, which has been shown to disrupt deoxyribonucleic (DNA) repair function in human tissue and experimental animal studies[13], and increase generation of carcinogenic reactive oxygen species[14].

Our findings with respect to dairy-related foods and nutrients are consistent with those from the most recent WCRF review which judged that dairy products (including evidence for total dairy, milk, and cheese, as well as dietary calcium) and calcium supplements probably decrease the risk of colorectal cancer[4]. Of the dairy-related foods and nutrients examined in the present study, all were inversely associated with risk of colorectal cancer, except for cheese and ice-cream. Our findings specifically for calcium (17% lower risk per 300 mg/day) and dairy milk (14% lower risk per 200 g/day), are larger in magnitude than those reported in the 2018 WCRF dose-response meta-analysis (9% and 6%, respectively for the same increments). In subsequent studies, a diet-wide analysis in

**Table 3 | Associations of FDR-significant dietary factors with risk of colorectal cancer, further adjusted for calcium, dairy milk, fruit, and wholegrains**

| Food or nutrient from diet only | Trend Increment | RR (95% CI) [1] Main model | RR (95% CI) [1] + Calcium added | RR (95% CI) [1] + Dairy milk added | RR (95% CI) [1] + Fruit added | RR (95% CI) [1] + Wholegrains added |
|---|---|---|---|---|---|---|
| Alcohol | 20 g/day | 1.15 (1.09,1.20) | 1.12 (1.07,1.18) | 1.13 (1.07,1.18) | 1.12 (1.07,1.17) | 1.14 (1.09,1.20) |
| Calcium | 300 mg/day | 0.83 (0.77,0.89) | - | 0.86 (0.79,0.95) | 0.88 (0.81,0.97) | 0.84 (0.78,0.90) |
| Dairy milk | 200 g/day | 0.86 (0.81,0.92) | 0.94 (0.86,1.02) | - | 0.86 (0.80,0.92) | 0.87 (0.82,0.93) |
| Phosphorus | 300 mg/day | 0.84 (0.78,0.91) | 0.95 (0.85,1.05) | 0.89 (0.82,0.98) | 0.92 (0.84,1.01) | 0.88 (0.81,0.95) |
| Riboflavin | 1 mg/day | 0.83 (0.75,0.91) | 0.96 (0.85,1.09) | 0.91 (0.81,1.03) | 0.95 (0.84,1.07) | 0.86 (0.78,0.95) |
| Magnesium | 100 mg/day | 0.84 (0.77,0.92) | 0.83 (0.83,1.01) | 0.88 (0.80,0.97) | 0.93 (0.84,1.04) | 0.80 (0.99,4.69) |
| Wholegrains | 20 g/day | 0.90 (0.85,0.95) | 0.91 (0.86,0.96) | 0.90 (0.85,0.96) | 0.92 (0.87,0.98) | - |
| Yogurt | 50 g/day | 0.92 (0.88,0.96) | 0.96 (0.91,1.00) | 0.92 (0.88,0.96) | 0.93 (0.89,0.98) | 0.93 (0.89,0.97) |
| Folate | 100 µg/day | 0.88 (0.82,0.95) | 0.92 (0.86,1.00) | 0.91 (0.84,0.98) | 0.94 (0.87,1.02) | 0.90 (0.83,0.97) |
| Carbohydrates | 50 g/day | 0.89 (0.83,0.96) | 0.92 (0.86,0.98) | 0.91 (0.85,0.97) | 0.93 (0.87,1.00) | 0.91 (0.85,0.98) |
| Total sugars | 50 g/day | 0.88 (0.81,0.95) | 0.92 (0.85,1.00) | 0.90 (0.83,0.98) | 0.94 (0.86,1.03) | 0.88 (0.82,0.96) |
| Red/processed meat | 30 g/day | 1.08 (1.03,1.12) | 1.06 (1.01,1.11) | 1.07 (1.03,1.12) | 1.06 (1.02,1.11) | 1.06 (1.02,1.11) |
| Fruit | 200 g/day | 0.90 (0.85,0.96) | 0.92 (0.86,0.98) | 0.90 (0.84,0.96) | - | 0.92 (0.86,0.98) |
| Vitamin C | 100 mg/day | 0.90 (0.83,0.96) | 0.91 (0.84,0.97) | 0.88 (0.82,0.95) | 0.92 (0.85,1.01) | 0.91 (0.84,0.98) |
| Breakfast cereal | 40 g/day | 0.93 (0.89,0.98) | 0.95 (0.90,1.00) | 0.95 (0.91,1.00) | 0.96 (0.91,1.01) | 0.95 (0.91,1.00) |
| Fibre | 5 g/day | 0.92 (0.86,0.97) | 0.93 (0.87,0.98) | 0.92 (0.86,0.97) | 0.96 (0.89,1.03) | 0.97 (0.90,1.04) |
| Potassium | 1000 mg/day | 0.89 (0.82,0.97) | 0.96 (0.88,1.05) | 0.94 (0.86,1.02) | 1.01 (0.91,1.11) | 0.91 (0.84,0.99) |

[1]Associations between each of the 17 foods or nutrients and colorectal cancer incidence calculated separately using Cox proportional hazards regression models that were stratified by year of birth, date of completion of the dietary survey (which is the baseline for this study), and region of residence (10 geographical regions: 9 in England and 1 in Scotland), and adjusted for area-based deprivation (fifths, based on the Townsend deprivation score, unknown), highest educational qualification (none, technical, secondary, tertiary, unknown), body mass index (<20, 20–22.49, 22.5–24.9, 25.0–27.49, 27.5–29.9, 30–32.49, 32.5–34.9, 35+ kg/m$^2$, unknown), height (<160, 160–164.9, ≥165 cm, unknown), strenuous exercise (none, ≤ once per week, > once per week, unknown),dietary energy intake (except for the analysis of energy and risk; fifths, unknown), alcohol (none, 1–5, 6–10, ≥ 11 drinks per week, unknown), smoking (never, past, current 1–4, current 5–9, current <10, current 10–14, current 15–19, current 20–24, current 25–29, current ≥ 30 cigarettes per day, unknown), current use of hormonal therapy for menopause (no, yes, unknown), and family history of bowel cancer (no, yes). For each dietary factor, we created a continuous variable using the re-measured mean intakes for each baseline category. Log-linear trends in risk across categories of baseline intakes were then calculated using the listed increments.

**Table 4 | Associations of sources of dietary calcium with colorectal cancer risk**

| Calcium source | Cases | RR (95% CI) [1] |
|---|---|---|
| **Dairy sources, by quintiles of intake**[2] | | |
| 1 | 2577 | 1 |
| 2 | 2465 | 0.93 (0.88,0.99) |
| 3 | 2425 | 0.90 (0.85,0.96) |
| 4 | 2432 | 0.90 (0.85,0.95) |
| 5 | 2352 | 0.86 (0.81,0.92) |
| **Non-dairy sources, by quintiles of intake**[3] | | |
| 1 | 2403 | 1 |
| 2 | 2482 | 1.00 (0.94, 1.06) |
| 3 | 2467 | 0.97 (0.91, 1.04) |
| 4 | 2457 | 0.95 (0.89, 1.02) |
| 5 | 2442 | 0.94 (0.86, 1.01) |
| *P for heterogeneity* | | 0.21 |

Mean daily intakes taken from the mean of the 24-hour dietary assessments. [1]Associations between calcium from dairy or non-dairy sources and colorectal cancer incidence calculated separately using Cox proportional hazards regression models that were stratified by year of birth, date of completion of the dietary survey (which is the baseline for this study), and region of residence (10 geographical regions: 9 in England and 1 in Scotland), and adjusted for area-based deprivation(fifths, based on the Townsend deprivation score, unknown), highest educational qualification (none, technical, secondary, tertiary, unknown), body mass index (<20, 20–22.49, 22.5–24.9, 25.0–27.49, 27.5–29.9, 30–32.49, 32.5–34.9, 35+ kg/m$^2$, unknown), height (<160, 160–164.9, ≥165 cm, unknown), strenuous exercise (none, ≤ once per week, > once per week, unknown), dietary energy intake (except for the analysis of energy and risk; fifths, unknown), alcohol (none, 1-5, 6-10, ≥ 11 drinks per week, unknown), smoking (never, past, current 1–4, current 5–9, current <10, current 10–14, current 15–19, current 20–24, current 25–29, current ≥30 cigarettes per day, unknown), current use of hormonal therapy for menopause (no, yes, unknown), and family history of bowel cancer (no, yes). [2]Further adjusted for quintiles of calcium intake from non-dairy sources. [3]Further adjusted for quintiles of calcium intake from dairy sources.

the EPIC study (~5000 cases among 387,000 participants) found 7% and 5% lower risks for the same increments[5], a study in the Nurses' Health Study II (349 cases among 94,000 participants) found a 15% lower risk for calcium per 300 mg/day[15], and a UK Biobank study (~2600 cases among 476,000 participants)[16] reported a 14% lower risk per 200 ml dairy milk/day (although this was not formally statistically significant, p for trend 0.07). One study in the China Kadoorie Biobank (3350 cases among 510,146 participants), with a much lower dairy intake than in western cohorts, found a suggestive positive association between dairy intake (largely coming from dairy milk) and colorectal cancer risk (eight percent higher risk per 50 g/day)[17]; it is possible that the association between dairy milk and colorectal cancer risk might differ in populations where a large majority cannot digest lactose, such as that in the China Kadoorie Biobank[18,19]. Our MR findings for genetically predicted milk intake in a European population provide evidence for a causal association of dairy and/or dietary calcium, adding to that from previous MR studies with similar findings based on far fewer colorectal cancer cases (i.e. ~7000 cases[20] and ~3400[21]) than our analysis (~53,000 cases). The MR findings for genetically predicted dairy milk were of much larger magnitude than what we observed in the observational analyses (40% versus 14% lower risk per 200 g/day), though genetically predicted intake represents the effect of lactase exposure throughout adult life, so this might be expected[22].

The associations we observed for dairy milk and the other dairy-related foods and nutrients with colorectal cancer are likely largely or wholly driven by calcium intake; this is based on the low *p*-value for the association between calcium and colorectal cancer risk, the large impact we found when adjusting the dairy-related food and nutrient associations with colorectal cancer for calcium, our residual-based analyses which showed that adding the estimated residuals for dairy milk intake given calcium did not independently add to the model, and

the investigation of the association of calcium and colorectal cancer risk by dietary source, which provided no evidence for heterogeneity of association with colorectal cancer risk by calcium source.

The probable protective role of calcium may relate to its ability to bind to bile acids and free fatty acids in the colonic lumen, thereby lowering their potentially carcinogenic effects[23,24]. Furthermore, experimental work in rats has shown that having higher levels of dietary calcium in the colonic lumen reduces colonic permeability, particularly if dietary phosphate levels are also high, thereby helping protect the intestinal mucosa from being injured by potentially harmful luminal contents (e.g. bile acids)[25]. Other experimental work suggests that calcium may also have direct effects on colonic tissue, for example, calcium may promote colorectal epithelial cell differentiation[26], enhance apoptosis, and reduce DNA oxidative damage in the colorectal mucosa[27]. Laboratory studies also suggest that dietary calcium may reduce the incidence of KRAS mutations in the colon[28]. The results from these previous experimental studies suggest that the potential protective effects of calcium appear to be related to its presence in the intestinal lumen. There is limited evidence on the role of circulating calcium in colorectal cancer risk, with the available genetic and observational evidence suggesting no clear association[15,29,30], though circulating concentrations of calcium are tightly regulated in the body and unlikely to be materially affected by moderate variations in dietary intake[31]. If the protective role of dairy milk and the other dairy-related foods is not wholly attributed to its calcium content, other possible mechanisms may relate to conjugated linoleic acid, butyric acid, and sphingomyelin which are present in dairy milk and have been shown to inhibit chemically-induced colon carcinogenesis in some animal models[32–36].

We could not investigate the association for calcium supplements in the present study. A recent meta-analysis of six cohort studies found that a 300 mg per day increase in calcium from supplements was associated with a 9% lower risk of incident colorectal cancer[37] but a randomised controlled trial in 36,282 postmenopausal women of supplementation with 1000 mg of elemental calcium (as calcium carbonate) with 40 μg of vitamin D3 daily for 7 years found no significant impact on risk[38]. However, mean calcium intakes in these women were relatively high at enrolment; the average intake from diet plus supplements was ~1100 mg/day, similar to the mean intake in the highest quintile of intake in the present study. It is therefore possible that the baseline calcium intakes in this trial were already high enough that the intervention with supplemental calcium had no further impact on colorectal cancer risk. Additionally, colorectal cancer has a long latency period, so it is possible that a follow-up period of seven years may have been insufficient to detect an effect of the intervention[39]. Apart from alcohol, the only dietary factor which was positively associated with colorectal cancer risk in these data was red and processed meat consumption. We found an 8% higher risk of colorectal cancer per 30 g/day higher red and processed meat consumption; this is equivalent to a 29% higher risk per 100 g/day, which is substantially larger than the 12% higher risk per 100 g/day reported in the 2018 WCRF dose-response meta-analysis[4]. This larger association might be partly explained by our use of repeat dietary intake measures to reduce the impact of measurement error and regression dilution bias. Similar to the WCRF report, we found a larger association for processed meat than for red meat, although the independent associations for red meat and processed meat separately were not robust to correction for multiple testing. However, in this paper we explored 97 dietary variables and corrected for multiple testing, and there are strong pre-existing hypotheses and evidence for some dietary factors, including for red and processed meat, and therefore correcting for multiple testing may have been a stringent approach for such dietary factors with consistent evidence of an association. Several mechanisms have been proposed to explain the positive associations observed for red and processed meat including haem iron, which may catalyse the formation of N-nitroso compounds that have been found to generate mutations in

the colon[40], cooking meat at high temperatures which forms heterocyclic amines and polycyclic aromatic hydrocarbons[41], and meat smoking or adding sodium nitrites or nitrates for preservation which can lead to the exogenous formation of N-nitroso compounds[41–43].

The magnitudes of the lower risks of colorectal cancer associated with greater intakes of breakfast cereal, fruit, wholegrains, carbohydrates, fibre, total sugars, folate, and vitamin C observed in this cohort were relatively small, and these inverse associations were affected by confounding by lifestyle factors and (except for fruit and wholegrains) by dietary factors. Suggested mechanisms for these inverse associations relate to wholegrains[44–46] and dietary fibre[4]. Wholegrains are a rich source of fibre and previous trial evidence shows that dietary fibre increases stool bulk; this leads to reduced transit time and dilutes the contents of the large bowel, thus possibly also diluting carcinogenic substances in bowel contents and the time such carcinogens are present in the colon[47]. Additionally, dietary fibre is fermented in the colon, forming short chain fatty acids such as butyrate, which reduce intestinal pH[48] and thus inhibit the conversion of primary bile acids into secondary bile acids, which promote cell proliferation[24]. It is also possible that other compounds found in these foods may have protective effects[4,49,50].

In this diet-wide prospective study on diet and colorectal cancer, we comprehensively investigated nearly 100 dietary factors in the same cohort, thereby reducing exposure selection bias, ensuring standardisation of confounding adjustment, and increasing the specificity of our findings[51]. We took a rigorous approach to explore the possibility of reverse causation by excluding women who reported changing their diet in the past 5 years due to illness from all the analyses, and in separate analyses by further restricting to women who self-reported good or excellent health at baseline, and by excluding the first 5 years of follow-up. We also assessed the potential role of confounding by assessing the impact of incremental adjustment for key confounders, and by conducting sensitivity analyses restricted to never smokers. The reproducibility and performance of the dietary assessment method used at baseline was assessed by comparison with records from 7-day food diaries[52]. In addition, we used a web-based 24-hour dietary questionnaire (the Oxford WebQ), validated against recovery biomarkers[53], to re-measure diet about 10 years later to estimate long-term diet and test for trends across baseline categories of intake[54]. In addition, the large sample size enabled us to look at proximal colon, distal colon and rectum separately. A limitation of our study was that for some dietary factors the range of re-measured (i.e. long-term) intakes across the extreme baseline groups was small, therefore for these factors we were limited in our ability to detect associations with disease. Also, we were unable to include some dietary items (e.g. butter) due to the format of the dietary survey. Additionally, although the women in the cohort are representative of middle-aged and older women living in the UK, they are predominantly of European ancestry. Therefore, the results are not necessarily generalisable to other populations, or to populations where a large majority cannot digest lactose (including e.g. many Asian populations).

In addition to confirming the well-established positive associations of alcohol, and red and processed meat consumption, with risk of colorectal cancer, this large prospective analysis provides robust evidence supporting the protective role of dietary calcium. Additional research is needed to investigate overall health benefits or risks associated with higher calcium intakes.

## Methods
### Ethical approval
The study was approved by the Oxford and Anglia Multi-Centre Research Ethics Committee and all participants gave written consent for follow-up through medical records. Further details of the study protocol and questionnaires have been published and the questionnaires can be viewed on the Million Women Study website[55,56].

## Study population

Between 1996 and 2001, 1.3 million women with a mean (SD) age of 56 (6) years who were invited to the National Health Service (NHS) Breast Screening Programme in England and Scotland joined the Million Women Study by completing the recruitment questionnaire, which collected information on demographic, lifestyle and social factors. Participants have been resurveyed at approximately 3–5 year intervals since recruitment, to update information on key exposures and to obtain additional information on new exposures of interest.

## Assessment of diet

The current analysis was based on the first resurvey (referred to as baseline) which was conducted around 3 years after recruitment (median year 2001, IQR 3) because this was the first questionnaire when women were asked about their dietary habits. This questionnaire asked participants about their diet during a typical week, including 130 quantitative or semi-quantitative questions on frequency of intake of specific foods and food groups (see Supplementary Methods). The mean daily intakes of nutrients were calculated by multiplying the frequency of consumption of each food or beverage by a specified portion size and the nutrient composition of that particular item. The short term repeatability of most of the diet questions was high, and comparison with estimates from 7-day diet diaries showed moderately good agreement (the median correlation for macronutrient intakes was 0.48, and for alcohol, calcium and fibre the correlations were 0.75, 0.62, and 0.62, respectively)[52].

Repeat measures of dietary intake were also derived from one web-based 24-hour dietary questionnaire (the Oxford WebQ) completed by a sub-sample (7%) of all women who completed the dietary questionnaire, on average ~10 years after baseline and before the end of follow-up[57].

In total, we included 97 dietary factors in our diet-wide analysis. Selection of foods and nutrients depended on their availability in both the dietary survey and the Oxford WebQ. Of the 97 selected foods and nutrients, 62 were measured quantitatively and 35 were measured as binary intakes. To enable calibrated estimates of intake to be made for all women for the 62 foods and nutrients that were quantitatively assessed, we first calculated the mean Oxford WebQ intake for each of the 62 dietary factors within each category (e.g. quintiles) of baseline intake for the sub-sample of women who completed the Oxford WebQ and then assigned these mean values to each baseline category for all women (see statistical analysis section for further details). Supplementary Data 6 presents the mean (SD) intakes for the quantitatively assessed foods and nutrients in women who completed one or more Oxford WebQs.

## Ascertainment of colorectal cancer

Participants were followed by electronic record linkage to routinely collected National Health Service (NHS) data on cancer registrations, deaths and emigrations, coded according to the International Classification of Diseases, 10[th] revision (ICD-10). The main endpoint for this study was incident colorectal cancer (ICD-10 C18-C20) and colorectal cancers were further classified as proximal (ICD-10 C18.0-C18.4), distal (C18.5-C18.7), or rectal (C19-C20) cancers.

## Exclusions and inclusions

In total, 866,535 women completed the baseline dietary questionnaire and had linked data for cancer and death. Of these, we excluded: 48,151 women with previous registration for malignant cancer (other than non-melanoma skin cancer, C44) or no follow-up before the dietary questionnaire completion date; 5090 women whose energy intake was outside the plausible range 2093-14,654 kJ per day (equivalent to 500–3500 kcal per day)[58]; 122,689 women who reported having changed their diet due to illness; and 147,827 women with missing data on any semi-quantitative dietary variables (including meat types, fish types, main carbohydrate sources, eggs, vegetables, fruit, sweets,

dairy, alcohol, and other beverages) leaving 542,778 women in the final analysis dataset (see Supplementary Figure 1 for participant flowchart).

## Statistical analysis

For 62 foods and nutrients for which there was a quantitative measure of intake we calculated trends in risk of colorectal cancer per increment in grams, milligrams or micrograms per day using information collected in the Oxford WebQ (in women who had not developed colorectal cancer at the time of completing this). To do this, dietary intake of these 62 foods and nutrients were first divided into categories, generally using quintiles; for foods with non-continuous distributions, we divided the dietary intakes into three to five categories using other appropriate cut-points to create approximately equal-sized groups based on the distribution of the data. We then derived repeat measures of intake within each baseline category by calculating the mean intakes for each food or nutrient category in women who had completed at least one 24-hour dietary assessment using the Oxford WebQ (based on the first completed Oxford WebQ where more than one had been completed), and assigning these mean intakes for all women in that category. These re-measured mean intakes for each baseline category were then treated as a continuous variable in order to calculate log-linear trends in risk across categories of baseline intakes. The trend analyses used the baseline categories of intake defined in all women, but assigned the mean intakes for each of these baseline categories using the Oxford WebQ intakes measured during follow-up; this method does not alter the baseline categories, or the HRs for each category compared to the reference category, but provides better estimates of long-term dietary intakes during follow up and therefore better estimates of trends in HRs associated with defined increments in dietary exposures (see Supplementary Methods for further description). Trend increments were selected based on the observed differences in WebQ derived intake between the lowest and highest baseline category (see increments for all in Supplementary Data 1). This approach reduces the impact of regression dilution bias and other forms of measurement error[54] and has previously been used for diet research in this cohort[59] and in the UK Biobank[16]. The remaining 35 foods (which included fruit and vegetable subtypes, ice cream, legumes, and soy milk) were divided into two categories of baseline intake ('weekly' vs 'less than weekly'). We checked the consistency of high versus low intakes between the baseline and re-measured intakes for these foods, and calculated risk of colorectal cancer for high versus low intakes using the baseline intakes.

We used Cox proportional hazards regression models to estimate hazard ratios (hereafter referred to as relative risks) and 95% confidence intervals for associations between each of the 97 dietary factors and colorectal cancer incidence separately. Person-years were calculated from the date when diet was reported up to whichever came first: diagnosis of cancer, emigration, death, or the end of follow up (31[st] December 2020). All analyses were stratified by year of birth, date of completion of the dietary survey, and region of residence (ten geographical regions: 9 in England and 1 in Scotland), and adjusted for area-based deprivation (fifths, based on the Townsend deprivation score at recruitment, unknown), highest educational qualification (none, technical, secondary, tertiary, unknown), body mass index ($< 20$, 20–22.49, 22.5–24.99, 25.0–27.49, 27.5–29.99, 30–32.49, 32.5-34.99, $\geq 35$ kg/m$^2$, unknown), height ($< 160$, 160–164.9, $\geq 165$ cm, unknown), strenuous exercise (none, $\leq$ once per week, $>$ once per week, unknown), dietary energy intake (except for the analysis of energy and risk; fifths, unknown), alcohol (except for the analyses of alcohol and risk; none, 1–5, 6–10, $\geq 11$ drinks per week, unknown), smoking (never, past, current 1–4, current 5–9, current <10, current 10–14, current 15–19, current 20–24, current 25–29, current $\geq 30$ cigarettes per day, unknown), current use of hormonal therapy for menopause (no, yes, unknown), and family history of large bowel cancer (no or unknown, yes). Data were missing for fewer than 5% of

women for each of the adjustment variables, with the exception of BMI (6.8% missing data); to ensure that the same women were being compared in all analyses the small number with a missing value for each particular variable were assigned to a separate category for that variable and included in the regression analysis. We used the Benjamini-Hochberg approach to calculate the False Discovery Rate (FDR) at 0.05 to account for multiple testing[60]. In total, 17 dietary factors met the FDR threshold and these factors were selected for further analyses.

## Further analyses

We calculated the pairwise Pearson's correlations between the 17 FDR-significant dietary factors to inform our assessment of the likely independence of their relationships with risk. We also examined the associations of these 17 FDR-significant dietary factors with risk of colorectal cancer by categories of intakes at baseline. To investigate the potential role of confounding by lifestyle factors in the associations between these 17 dietary factors and colorectal cancer, modelled as log-linear trends in risk across categories of baseline intakes, we calculated the change in the log relative risk associated with each of the 17 dietary factors after differing levels of adjustment for potential confounders. We also investigated the degree to which each of the 17 dietary associations were independent of the four dietary factors that were either the most strongly related to risk (calcium, dairy milk) or were foods that were substantially correlated with the other dietary factors (fruit, and wholegrains), by assessing the impact of further adjustment for each of these four factors individually.

## Dairy milk, and dietary calcium

To investigate the separate, independent associations of total dietary calcium and dairy milk with the risk of colorectal cancer we used the residuals method[61]. To do this, we first obtained the calcium and dairy milk residuals from two separate linear regressions: one regression of dietary calcium on dairy milk, and the other of dairy milk on dietary calcium. The dietary calcium and dairy milk residual values were divided into quintiles. We then compared the associations between dairy milk and colorectal cancer risk using the fully adjusted Cox regression models used in the main analysis with and without adding the quintiles of dietary calcium residuals using likelihood-ratio tests (LRT). The same analysis was repeated for the association between dietary calcium and colorectal cancer by adding and removing the dairy milk residual quintiles. We additionally investigated the associations of dietary calcium with risk of colorectal cancer according to whether it was derived from dairy sources or non-dairy sources and compared the associations using a test for heterogeneity.

## Sensitivity analyses

Given that early symptoms of colorectal cancer could plausibly lead to changes in diet many years before diagnosis, we conducted sensitivity analyses restricted to women in self-reported excellent or good health at baseline, and to risk of colorectal cancer in the period 5 or more years after baseline to assess the likely impact of potential reverse causation bias. We also examined potential differences in associations by cancer sub-site including proximal colon, distal colon and rectum. We further assessed associations between the 17 FDR-significant dietary factors and colorectal cancer stratified by smoking status, BMI, area-based deprivation and alcohol intake to investigate potential confounding and differences by strata. Chi squared tests were used to assess p for heterogeneity in associations by cancer sub-site and across strata for each potential lifestyle confounder.

## Mendelian randomisation (MR) using lactase polymorphism

Given the strong and consistent association of dairy products, dairy milk, and calcium with a lower risk for colorectal cancer in previous studies[4] and in the present analysis, we further assessed evidence of causality using MR. Dietary calcium intake does not have an established genetic variant to estimate causal associations, but dairy milk intake in populations of European ancestry is robustly predicted by the SNP rs4988235[62] located in the *MCM6* gene. This SNP is immediately upstream of the *LCT* gene that codes for the lactase enzyme necessary to digest the lactose in dairy milk, and the "lactase persistence" genotype is associated with persistence of intestinal lactase production into adulthood[63]. Dairy milk intake is a large contributor of calcium in European populations, with ~one third of all dietary calcium coming from dairy milk in the Million Women Study, so genetically predicted milk intake may also serve as an instrument for calcium intake. We conducted a two-sample MR using a Wald ratio to estimate the associations of SNP rs4988235 with risk for colorectal, colon and rectal cancers. We assigned each additional genetically predicted milk intake increasing allele an increment of 17.1 g/d of dairy milk based on findings from a European cohort study including ~21,900 participants[64], and then rescaled this increment to 200 g/d. Summary statistics for the associations of the LCT variant (rs4988235) with colorectal cancer were obtained from a GWAS of 99,152 participants (52,865 colorectal cancer cases and 46,287 controls). The GWAS data were from a meta-analysis within the ColoRectal Transdisciplinary Study, the Colon Cancer Family Registry, and the Genetics and Epidemiology of Colorectal Cancer consortium (GECCO), making this combined analysis the largest meta-analysis for colorectal cancer in adults of European ancestry. Imputation was performed using the Haplotype Reference Consortium r1.0 reference panel and the regression models were further adjusted for age, sex, genotyping platform (when appropriate), and genomic principal components[62].

All of the statistical analyses in the present study were performed using Stata statistical software 18.1 (StataCorp, College Station, TX) and R 4.1.

## Reporting summary

Further information on research design is available in the Nature Portfolio Reporting Summary linked to this article.

## Data availability

Information on data access is available at www.millionwomenstudy. org/data_access/.GECCO collaborators and consortium members can access the GECCO portal here: https://research.fredhutch.org/peters/ en/genetics-and-epidemiology-of-colorectal-cancer-consortium.html.

## Code availability

The code for the MR analysis in this study can be found: https://github. com/karlsmithbyrne/Lactase_MR/tree/main.

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

## Acknowledgements

The authors would particularly like to acknowledge the significant contribution of Professor Dame Valerie Beral for the initiation of the Million Women Study and for her expertise and guidance in this research. She was Chief Investigator of the Study until 2020 and commented on early versions of this manuscript before she died in August 2022. We thank the women who have participated in the Million Women Study as well as the staff from the participating NHS breast screening centres. This work uses data provided by patients and collected by the NHS as part of their care and support. We thank NHS England and Public Health Scotland for the health outcomes data. This work was funded by Cancer Research UK (C570/A16491 and A29186) and the UK Medical Research Council (MR/K02700X/1). KEB's work on this project was funded by a Girdlers' Health Research Council Fellowship (3716491). The funders had no role in study design, data collection and analysis, decision to publish or preparation of the manuscript. For the purpose of open access, the authors have applied a Creative Commons Attribution (CC BY) licence to any Author Accepted Manuscript version arising.

## Author contributions

These authors contributed equally: Keren Papier, and Kathryn E Bradbury. Study concept and design: K.P., K.E.B., V.B., T.J.K., and G.K.R. Statistical analysis: A.B. (observational data) and K.S.B. (genetic data). Drafting of initial manuscript: K.P. and K.E.B. Interpretation of the data, critical revision of the manuscript for important intellectual content, and approval of the final submitted version: K.P., K.E.B., A.B., I.B., K.S.B., M.J.G., S.I.B., L.L.M., A.H.W., U.P., V.B., T.J.K., and G.K.R.

## Competing interests

UP was a consultant with AbbVie and her husband is holding individual stocks for the following companies: BioNTech SE – ADR, Amazon, CureVac BV, NanoString Technologies, Google/Alphabet Inc Class C, NVIDIA Corp, Microsoft Corp. The remaining authors declare no competing interests.
