## [Transparent Peer Review file · Nature Communications]

Diet-wide analyses for risk of colorectal cancer: prospective study of 12,250 incident cases among 543,000 women in the UK

Corresponding Author: Dr Keren Papier

Version 0:

Reviewer comments:

Reviewer #1

(Remarks to the Author)

Papier and Bradbury et al. used a diet-wide approach to investigate the dietary risk factors for colorectal cancer leveraging the Million Women Study, combined with a Mendelian randomization analysis. This study is interesting and addresses a critical gap in research. Overall, the paper is well-written, with methods well-described and results well-organized. The authors also provided a comprehensive discussion of the potential mechanisms that could explain the observed associations. Below are some suggestions/questions that will help to improve this manuscript.

Abstract

1. Line 34: please be specific on how the 97 foods and nutrients were measured. Using food frequency questionnaires, 24-hour diet recalls, or something else?
2. Line 36: is the genetic analysis an ad hoc analysis? If so, please specify this in the abstract. You may consider mentioning something like: "Based on the results of diet-wide analyses, we further assessed xxx".
3. Results: may consider using $p < 0.0001$ to note the small p values.
4. Line 47: please specify what "further analysis" means here. What is the type of this analysis?
5. Lines 49-52: this sentence reads like a discussion point rather than a result description. Suggest rephrasing or removing it from the abstract.

Methods

1. Line 112: please be specific about the number of repeated measures. Was that one 24-hour dietary questionnaire or multiple 24-hour dietary measurements?
2. Line 115: It is not very clear if the 97 dietary factors can be captured in both the dietary survey and the Oxford WebQ. If only 7% of the participants completed the Oxford WebQ, how to ensure repeat measures of dietary intake data were available for all participants? Please add more details to help readers understand this better.
3. Exclusions and inclusions: it would be helpful to add a flowchart to help readers better understand the process of participant selection.
4. Line 129: please justify why use 2093-14654 kJ per day as the cut-off.
5. Lines 133-134: please provide a brief description of the population characteristics in the result section.
6. Line 155: please confirm if "women-years" is the proper terminology to use.
7. Line 175: please provide a rationale why to conduct the pairwise correlation analysis.
8. Lines 176-177: please be specific about how the intake at baseline was categorized. Using quintiles?
9. Line 203: suggest changing "residual confounding" to "potential confounding".
10. Line 213: please define MWS. It is not clear what this term means.

Results

1. Lines 236-237: suggest using $p < 0.0001$ to note the very small p values.
2. Line 250: please define "progressive adjustment" and confirm it is the correct term to use. Not exactly sure what it means in the context.
3. Line 251: would it be more appropriate to use "the most statistically significant association" than "the strongest association"? When using the word "strongest", it implies that the magnitude of the association is the largest.

4. Line 256 and lines 300 and 408 (discussion): suggest deleting the word “residual” or changing it to “potential”.
5. Line 282: is “phet” the p-value for interaction? Please confirm. Suggest changing it p-value for interaction, which is more commonly used in epidemiological studies.
6. Lines 283-285: are these p-values corresponding to strata p-values? What are the p-values for interactions? Are they statistically significant?

Table 3 and Table 4, footnote: not exactly sure what it means by saying “regression models stratified by year of birth, date of completion of the 3-year xxx”. Are these stratification analyses? Please confirm and revise accordingly.

(Remarks on code availability)

Reviewer #2

(Remarks to the Author)

In this investigation, the authors leverage a large data resource to examine associations of 97 dietary factors with CRC risk. Although it is well appreciated that aspects of diet are likely to impact CRC risk, there is a lack of consensus around several specific dietary factors, including calcium intake. Thus, the authors contribution with this very robust sample size is a valuable one. Overall, the study adds to the body of evidence on the association of dietary factors with CRC, particularly for calcium, dairy milk, and alcohol. However, there are some methodologic points that merit closer examination.

Major comments:

- Potentially relevant associations with dietary factors other than calcium, dairy milk, and alcohol consumption are downplayed. In particular, the authors state in the abstract that “breakfast cereal, fruit, whole grains, ... were inversely associated with risk, though may have been influenced by residual confounding by lifestyle and other dietary factors.” The authors found other nutrients/diet factors in addition to milk and calcium, but their associations were mostly attenuated by adjusting for milk, calcium, fruit or whole grains (paragraph starting with line 258). However, fruit and whole grain inverse associations themselves were not attenuated by adjusting for the other dietary factors (Table 3). These associations are important, in addition to calcium/dairy, and seem somewhat downplayed in the abstract and discussion, which in turn could downplay the importance of overall diet quality vs single nutrients or single foods.
- The Mendelian Randomization analysis does not add to the paper as presented. The methods for this analysis are not well described (e.g., no sample size is provided) and the results are not presented in any table or supplement. As presented, this analysis does not add meaningfully to the manuscript. There is also some question as to whether genetically predicted milk intake could really be a proxy for calcium intake when it only accounts for about 1/3 of the womens' calcium intake?
- Methods for parameterizing / classifying dietary variables for analysis is not clear. In particular, the authors talk about classifying quantiles of the baseline dietary variables for analysis, but present results for continuous models. The use of the 10-year follow-up data is also unclear, and it is unclear whether the small subset of participants included in the follow-up is representative and/or how diet reporting for this subset changed over the 10-year period.
- Analyses to address possible reverse causality are not clearly motivated. In particular, the authors conduct a sensitivity analysis excluding the first 5-years of follow-up, but it is unclear why they chose such a long interval of exclusion.

Minor Comments:

- It is not clear how the authors decided which of the 17 dietary factors in Table 3 to use as adjustment factors (other than milk and calcium). Perhaps fruit and whole grains were selected because are the only two “whole” food groups remaining? Clarity regarding this choice is needed.
- Relative risk (RR) is used to refer to hazard ratio (HR) for most of the paper, but HR is used in a couple of places (Figure 1 and Supplementary Methods). It may be helpful to refer to HR first and then “hereafter referred to as relative risks”, or similar. Whichever terminology is used (RR or HR) should be consistent throughout the paper.
- Line 186: Citation for residual method eventually goes back to Willett and Stampfer 1986 <https://academic.oup.com/aje/article/124/1/17/150019> (it seems like the paper cited here just used the method). Relatedly, line 189: why use quintiles and not just the residuals themselves (noted food intake values are divided into quintiles for some analyses, but the main findings in Table 3 are reported by trend increment)?
- Line 337-338: “based on the low p-value for the association....” The low p-value could be due to large sample size, not necessarily reflective of the importance of the finding. In study of such large size, the authors should be cautious to interpret p-values outside the context of the magnitude of association.
- Line 343: Having “The probable protective...” start a new paragraph may improve readability.
- Figure 1: what are the units for the foods/nutrients in the DWAS? As shown in Table 3?

(Remarks on code availability)

Reviewer #3

(Remarks to the Author)

(Remarks on code availability)

Version 1:

Reviewer comments:

Reviewer #1

(Remarks to the Author)

Thank you for the opportunity to review this revised manuscript. The authors have addressed most of the prior comments. However, a major concern remains regarding the methods used to estimate food and nutrient intake – it is still not clear how the intake was estimated. Additionally, it is questionable whether the method used is valid for estimating “long-term” or “usual intake.”

Specifically, the authors stated in their response to my question #2 (Methods) that they “calculated mean intakes of the 97 dietary factors from one web-based 24-hour dietary questionnaire.” However, based on Supplementary Table 1, the total number of dietary factors reported is 61, and some of these food items differ from those presented in Supplementary Table 2. It is confusing how the 97 dietary factors could be calculated using the WebQ method, which only captured 61 foods and nutrients.

Lines 122-123 state: “Selection of foods and nutrients depended on their availability in both the dietary survey and the Oxford WebQ.” Does this mean that only the overlapping food items or nutrients were included in the analysis? If so, it would be very helpful if the authors could show (perhaps using a Venn diagram) the number of overlapping food items measured using the two different dietary assessment tools.

Lines 126-130 stated: “To enable calibrated estimates of intake to be made for all women, we first calculated the mean Oxford WebQ intake for each of the 97 dietary factors within each category of baseline intake for the sub-sample of women who completed the Oxford WebQ and then assigned these mean values to each baseline category for all women (see statistical analysis section for further details).” Would it be clearer to add in parentheses what the “category” refers to since “category” can also mean a category of food subgroups? For example, “...within each category (e.g., quintiles)...?”

Lines 149-151 stated: “...we divided the dietary intakes into three to five categories using other appropriate cut-points.” Are there any justifications for choosing these cut points?

The authors mentioned that they used “a statistical approach to estimate ‘usual intake’ during follow-up.” Are there any references to support the validation of this approach to estimate “long-term” or “usual intake”? If not, I suggest the authors tone it down. It would be more appropriate to maintain transparency by clearly describing the methods used to obtain the estimations. Please also revise the abstract accordingly.

Regarding Table 3, please specify how the exposure was modeled. Was it modeled as a category variable based on the baseline categories? Or was it modeled as a continuous variable using the increment defined in column 2 “trend increment”? Please also make sure this is clearly described in the “statistical method” section.

(Remarks on code availability)

Reviewer #2

(Remarks to the Author)

The authors have been responsive to comments from the previous review. I have no further comments to provide.

(Remarks on code availability)

Reviewer #3

(Remarks to the Author)

(Remarks on code availability)

Version 2:

Reviewer comments:

Reviewer #1

(Remarks to the Author)

Thank you for the opportunity to review the revised manuscript. The authors have adequately addressed all of my comments.

(Remarks on code availability)

Response to reviewers

Re: Diet-wide analyses for risk of colorectal cancer: prospective study of 12,250 incident cases among 543,000 women in the UK

We thank the reviewers for their comments, these have greatly improved our manuscript. We have addressed the reviewers' comments point-by-point below. We present reviewers' comments below, followed by our responses (indented). Revised manuscript text appears as red text.

Reviewer reports	2
Reviewer #1	2
Reviewer #2	9
Reviewer #3	14
References.....	15

Reviewer reports

Reviewer #1

Papier and Bradbury et al. used a diet-wide approach to investigate the dietary risk factors for colorectal cancer leveraging the Million Women Study, combined with a Mendelian randomization analysis. This study is interesting and addresses a critical gap in research. Overall, the paper is well-written, with methods well-described and results well-organized. The authors also provided a comprehensive discussion of the potential mechanisms that could explain the observed associations. Below are some suggestions/questions that will help to improve this manuscript.

Abstract

1. Line 34: please be specific on how the 97 foods and nutrients were measured. Using food frequency questionnaires, 24-hour diet recalls, or something else?

Authors' Response:

We thank the reviewer for their suggestion and have now revised the abstract text with more specific information.

Abstract, p. 2, Lines 36-41

Methods: We examined associations of 97 foods and nutrients with risk of colorectal cancer in 542,778 women enrolled in the Million Women Study using multivariable-adjusted Cox regression models. **Women were firstly categorised according to their intake of these foods and nutrients on the basis of a validated short food frequency questionnaire which was completed by all women in 2000-2004. Long-term intake within each baseline category was then estimated using information from a web-based 24-hour dietary assessment (conducted 10 years later, on average) which was available in a large subset of women, to reduce the impact of measurement error and allow for changes in intake over time.**

2. Line 36: is the genetic analysis an ad hoc analysis? If so, please specify this in the abstract. You may consider mentioning something like: "Based on the results of diet-wide analyses, we further assessed xxx".

Authors' Response:

We agree that this would be helpful to clarify and have now added this in our revised text.

Abstract, p. 2, Line 41

Based on the results of the diet-wide analyses, we further assessed the association between genetically predicted milk intake (**which may also act as a proxy of calcium intake**) and colorectal cancer risk in the ColoRectal Transdisciplinary Study, the Colon Cancer Family Registry, and the Genetics and Epidemiology of Colorectal Cancer Consortium (GECCO) using a two-sample Mendelian randomisation (MR) analysis.

3. Results: may consider using $p < 0.0001$ to note the small p values.

Authors' Response:

We would prefer to specify the small p values for calcium and alcohol to highlight the strength of the association with risk of colorectal cancer for these two dietary factors.

4. Line 47: please specify what “further analysis” means here. What is the type of this analysis?

Authors' Response:

We have now further described this in the revised text.

Abstract, p. 2, Lines 53-54

Other dairy-related factors including dairy milk, yogurt, riboflavin, magnesium, phosphorus, and potassium were also inversely associated with colorectal cancer risk, though further analysis (which included controlling for calcium, residual-based analyses, and investigations by dietary calcium source) showed that confounding by calcium intake was likely to account for these associations.

5. Lines 49-52: this sentence reads like a discussion point rather than a result description. Suggest rephrasing or removing it from the abstract.

Authors' Response:

We agree and have now deleted this description.

Methods

1. Line 112: please be specific about the number of repeated measures. Was that one 24-hour dietary questionnaire or multiple 24-hour dietary measurements?

Authors' Response:

Thank you for your comment. We have now clarified this in our revised text.

Methods, p. 4, Lines 121-122

Repeat measures of dietary intake were also derived from one web-based 24-hour dietary questionnaire (the Oxford WebQ) completed by a sub-sample (7%) of all women who completed the dietary questionnaire, on average ~10 years after baseline and before the end of follow-up [1].

2. Line 115: It is not very clear if the 97 dietary factors can be captured in both the dietary survey and the Oxford WebQ. If only 7% of the participants completed the Oxford WebQ, how to ensure repeat measures of dietary intake data were available for all participants? Please add more details to help readers understand this better.

Authors' Response:

We assessed intakes of 97 dietary factors in the baseline dietary survey for all women. Then, to reduce the impact of measurement error at baseline and allow for changes in intake over time, we used a statistical approach that was developed using repeat measurements in a subset of the cohort to estimate “usual intakes” during follow-up, in the exposure categories defined at baseline for all 97 dietary factors. In nutritional epidemiology, repeat measures of diet (using the same or an alternative assessment tool) in a relatively small subsample are commonly used to improve the reliability and repeatability of intakes for all participants [14]. To do this in the present study, we calculated mean intakes of the 97 dietary factors from one web-based 24-hour dietary questionnaire (the Oxford WebQ) completed by a sub-sample (7%, n=36,597) and then assigned these mean intakes to the baseline dietary questionnaire intake categories defined for all participants; thus for the nutritional factors categorised in quintiles, for example, the mean values assigned for each quintile were based on ~37 thousand women, providing reliable estimates of usual long-term intakes. This approach doesn't alter the RRs for each category but instead may slightly modify the magnitude of any dose response estimate.

We have now clarified this in our revised text.

Methods, p. 4, Lines 126-130

In total, we included 97 dietary factors in our diet-wide analysis. Selection of foods and nutrients depended on their availability in both the dietary survey and the Oxford WebQ. **To enable calibrated estimates of intake to be made for all women, we first calculated the mean Oxford WebQ intake for each of the 97 dietary factors within each category of baseline intake for the sub-sample of women who completed the Oxford WebQ and then assigned these mean values to each baseline category for all women (see statistical analysis section for further details).** Supplementary Table 1 presents the mean (SD) intakes for the quantitatively assessed foods and nutrients in women who completed one or more Oxford WebQs.

3. Exclusions and inclusions: it would be helpful to add a flowchart to help readers better understand the process of participant selection.

Authors' Response:

We agree and have now added this as a new Supplementary Figure 1.

4. Line 129: please justify why use 2093-14654 kJ per day as the cut-off.

Authors' Response:

The selected cut offs are commonly used to assess plausible self-reported dietary intakes for relatively active women. We have now added the reference for these cut offs in our revised text. We have now also described the kilocalorie equivalents from which these were originally derived.

Methods, p. 4, Lines 143-144

In total 866,535 women completed the baseline dietary questionnaire and had linked data for cancer and death. Of these, we excluded: 48,151 women with previous registration for malignant cancer (other than non-melanoma skin cancer, C44) or no follow-up before the dietary questionnaire completion date; 5,090 women whose energy intake was outside the plausible range 2093-14,654 kJ per day (equivalent to 500-3500 kcal per day) [14];

14. Willett, W. *Nutritional Epidemiology*; Oxford University Press: Oxford, UK, 2012

5. Lines 133-134: please provide a brief description of the population characteristics in the result section.

Authors' Response:

We have now added a brief description in our revised text.

Results, p. 7, Lines 259-262

These 542,778 women had a mean (SD) of 16.6 (4.8) years of follow-up, during which 12,251 women were diagnosed with incident colorectal cancer. Table 1 shows participants' characteristics overall and according to whether they developed incident colorectal cancer during follow-up. Women who developed colorectal cancer were older, taller, had more family history of bowel cancer, and had more adverse health behaviours compared with participants overall.

6. Line 155: please confirm if “women-years” is the proper terminology to use.

Authors' Response:

We have changed the term to person-years in our revised text.

Methods, p. 5, Line 177

Person-years were calculated from the date when diet was reported up to whichever came first...

7. Line 175: please provide a rationale why to conduct the pairwise correlation analysis.

Authors' Response:

We have now described this in our revised text.

Methods, p. 5, Lines 197-198

We calculated the pairwise correlations between the 17 FDR-significant dietary factors to inform our assessment of the likely independence of their relationships with risk.

8. Lines 176-177: please be specific about how the intake at baseline was categorized. Using quintiles?

Authors' Response:

We have now described this in more detail in our revised text.

Methods, p. 4, Lines 154-155

To do this all baseline dietary intakes were first divided into categories, generally using quintiles; for **foods with non-continuous distributions we divided the dietary intakes into three to five categories using other appropriate cut-points.**

9. Line 203: suggest changing “residual confounding” to “potential confounding”

Authors’ Response:

Thank you for this suggestion we have edited this accordingly in our revised text.

Methods, p. 6, Line 229

We further assessed associations between the 17 FDR-significant dietary factors and colorectal cancer stratified by smoking status, BMI, area-based deprivation and alcohol intake to investigate **potential** confounding and differences by strata.

10. Line 213: please define MWS. It is not clear what this term means.

Authors’ Response:

Thank you for picking this up. We have now spelled the study name in full.

Methods, p. 6, Line 241

Dairy milk intake is a large contributor of calcium in European populations, with ~one third of all dietary calcium coming from dairy milk in the **Million Women Study, ...**

Results

1. Lines 236-237: suggest using $p < 0.0001$ to note the very small p values.

Authors’ Response:

We prefer to list the very small p values for calcium and alcohol in full to demonstrate the strength of the association with risk of colorectal cancer compared to the other 15 FDR associations.

2. Line 250: please define “progressive adjustment” and confirm it is the correct term to use. Not exactly sure what it means in the context.

Authors’ Response:

We agree that this term may not be common and have revised our text.

Results, p. 7, Line 281-285

Sequential adjustment for different levels of potential lifestyle confounders across the models did not materially change the log relative risks for **those dietary factors which**

showed the most statistically significant associations (based on lowest p values) with risk of colorectal cancer (alcohol, calcium, dairy milk).

3. Line 251: would it be more appropriate to use “the most statistically significant association” than “the strongest association”? When using the word “strongest”, it implies that the magnitude of the association is the largest.

Authors’ Response:

We agree and have revised this. See response above for revised text.

4. Line 256 and lines 300 and 408 (discussion): suggest deleting the word “residual” or changing it to “potential”.

Authors’ Response:

We have now deleted the word residual in line 430 (previously 408) but prefer to keep it in lines 277 (previously 256) and 321 (previously 300) since further adjustment led to attenuation of the magnitude of some of the associations. Given that we see this attenuation when adjusting for variables which were likely measured with some error, there will likely be even greater attenuation if these were perfectly measured, suggesting that that there may be residual confounding.

5. Line 282: is “phet” the p-value for interaction? Please confirm. Suggest changing it p-value for interaction, which is more commonly used in epidemiological studies.

Authors’ Response:

The term phet here denotes the p-value arising from a test of heterogeneity between estimates of a given association within distinct strata defined by various characteristics. Although this is essentially equivalent to a test for interaction by subgroup or cancer subsite, the test does not strictly relate to the significance of an interaction term so we would prefer to stick to the term “p-value for heterogeneity”. We have now described the test and the phet in full in our revised text.

Methods, p. 6, Lines 229-231

Chi squared tests were used to assess p for heterogeneity in associations by cancer subsite and across strata for each potential lifestyle confounder.

Results, p. 8, Lines 313-316

We also found no significant heterogeneity in the observed associations for the FDR-significant dietary factors by cancer sub-site, except for alcohol, which appeared to be less detrimental in the proximal colon and most harmful in the rectum (Supplementary Table 5; p for heterogeneity by subsite=0.02).

Supplementary material, Supplementary Table 5 and Supplementary Table 6 footnotes

Phet stands for p for heterogeneity

6. Lines 283-285: are these p-values corresponding to strata p-values? What are the p-values for interactions? Are they statistically significant?

Authors' Response:

Please refer to our response in the question above.

Table 3 and Table 4, footnote: not exactly sure what it means by saying “regression models stratified by year of birth, date of completion of the 3-year xxx”. Are these stratification analyses? Please confirm and revise accordingly.

Authors' Response:

These footnotes are describing the adjustment factors that were added in as strata in the Cox regression model. To further clarify we tried to describe this similarly to how the adjustments were described in our revised text.

Tables 3 and 4, p. 19-20, Footnotes

¹Associations between each of the 17 foods or nutrients and colorectal cancer incidence calculated separately using Cox proportional hazards regression models that were stratified by year of birth, date of completion of the dietary survey (which is baseline for this study), and region of residence (10 geographical regions (9 in England, and Scotland), and adjusted for ...

Reviewer #2

In this investigation, the authors leverage a large data resource to examine associations of 97 dietary factors with CRC risk. Although it is well appreciated that aspects of diet are likely to impact CRC risk, there is a lack of consensus around several specific dietary factors, including calcium intake. Thus, the authors contribution with this very robust sample size is a valuable one. Overall, the study adds to the body of evidence on the association of dietary factors with CRC, particularly for calcium, dairy milk, and alcohol. However, there are some methodologic points that merit closer examination.

Major comments:

1. Potentially relevant associations with dietary factors other than calcium, dairy milk, and alcohol consumption are downplayed. In particular, the authors state in the abstract that “breakfast cereal, fruit, whole grains, ... were inversely associated with risk, though may have been influenced by residual confounding by lifestyle and other dietary factors.” The authors found other nutrients/diet factors in addition to milk and calcium, but their associations were mostly attenuated by adjusting for milk, calcium, fruit or whole grains (paragraph starting with line 258). However, fruit and whole grain inverse associations themselves were not attenuated by adjusting for the other dietary factors (Table 3). These associations are important, in addition to calcium/dairy, and seem somewhat downplayed in the abstract and discussion, which in turn could downplay the importance of overall diet quality vs single nutrients or single foods.

Authors' Response:

We thank the reviewer for picking this up and agree that the associations for fruit and wholegrains were not substantially affected by adjustment for other dietary factors, as was the case for other inversely associated foods and nutrients. However adjustment for lifestyle characteristics did lead to an attenuation in the magnitudes of associations for fruit and wholegrains, suggesting that these associations may have been influenced by residual confounding by non-dietary lifestyle factors. To ensure we don't downplay these two results while also providing context for the strength of these associations, we have now revised the abstract and the discussion text accordingly. Please see our revised text below.

Abstract, p. 2, Lines 58-59

Of the remaining dietary factors that were associated with colorectal cancer risk, only red and processed meat intake was associated with increased risk. Breakfast cereal, fruit, wholegrains, carbohydrates, fibre, total sugars, folate, and vitamin C were inversely associated with risk (all $p < 0.01$) but these associations were attenuated after adjustment for lifestyle factors. ~~though these associations may have been influenced by residual confounding by lifestyle and other dietary factors~~

Discussion, p. 8, Line 337

In addition, we observed inverse associations with risk for breakfast cereal, fruit, wholegrains, carbohydrates, fibre, total sugars, folate, and vitamin C, but these inverse associations may have been influenced by residual confounding by lifestyle and/or other dietary factors.

Discussion, p. 10, Line 431

The magnitudes of the lower risks of colorectal cancer associated with greater intakes of breakfast cereal, fruit, wholegrains, carbohydrates, fibre, total sugars, folate, and vitamin C observed in this cohort were relatively small, and these inverse associations were affected by confounding by lifestyle factors and, **except for fruit and wholegrains,** by dietary factors.

2. The Mendelian Randomization analysis does not add to the paper as presented. The methods for this analysis are not well described (e.g., no sample size is provided) and the results are not presented in any table or supplement. As presented, this analysis does not add meaningfully to the manuscript. There is also some question as to whether genetically predicted milk intake could really be a proxy for calcium intake when it only accounts for about 1/3 of the womens' calcium intake?

Authors' Response:

We thank the reviewer for their comments. However, we feel that our Mendelian randomisation (MR) analyses do add value to the prospective analyses in this manuscript. Finding concordant evidence from two different study designs and methods (observational and genetic) that have different sources of bias and confounding helps investigate the potential role of exposures in the aetiology of disease more confidently. For diet, there are a limited number of robust genetic instruments that can be used to assess concordant results. Of those available, alcohol and genetically predicted dairy milk (which is a measure of lactase exposure throughout life) are the most robust. Given that alcohol is already a well-established risk factor for colorectal cancer, we sought to assess the role of genetically predicted dairy milk in colorectal cancer risk. To do so, our analyses were conducted using a genetic variant that lies adjacent to the gene that codes for the lactase enzyme required to digest lactose, which is an established and biologically plausible instrumental variable for milk intake, and the world's largest GWAS of colorectal cancer. Our MR results provide strong support for a protective role of dairy milk intake in colorectal, colon and rectal, cancer incidence, which is directionally and quantitatively concordant with our findings based on dietary intake of milk. While it is correct that only approximately one third of the calcium consumed in this population is from dairy milk, the difference in milk intake predicted by the lactase SNP, of 17 g/d per allele, would be expected to result in a difference in calcium intake of ~20 g/d (based on the average calcium content of milk of ~120 mg/100 g). Moreover, genetically predicted milk intake represents the effect of lactase exposure throughout adult life and may therefore be a better marker of long-term intake of milk, and therefore calcium, compared with that based on self-reported intake in middle-age. In addition to providing concordant evidence for colorectal cancer, we also present similar novel concordant results specifically for rectal and colon cancer sub-sites, which we think merit presentation. We agree with the reviewer that it is difficult to be entirely clear about whether the association we see for colorectal cancer is principally explained by dairy milk or calcium and have now described this in our revised text.

Methods, p. 6, Lines 241-242

Dairy milk intake is a large contributor of calcium in European populations, with ~one third of all dietary calcium coming from dairy milk in the **Million Women Study**, so genetically predicted milk intake **may also serve as an instrument for calcium intake**.

Discussion, p. 8, Lines 325-330

In this single largest prospective study of diet and colorectal cancer, we found a marked positive association for alcohol, and a strong inverse association for calcium. **Inverse associations were also observed with other dairy-related factors including dairy milk, yogurt, riboflavin, magnesium, phosphorus, and potassium which, on further analysis, appeared to be primarily due to the association of these dietary factors with calcium. Further evidence for a potentially causal role for calcium in CRC incidence was provided by an accompanying analysis of genetically-predicted milk intake, which is likely to also reflect calcium intake.**

On description of the method

We acknowledge that in many MR studies, where multiple genetic variants are used, the reporting of several methods of analysis can be helpful such as methods that provide estimates of the central tendency of single variant Wald ratios (beta for the outcome/beta for the exposure). However, in our study we used only one genetic variant with high biological plausibility [2] and therefore the description in the methods section comprehensively covers analyses for the single variant. The sample size for the GWAS of dairy milk and colorectal cancer is described in the Methods section line 246.

3. Methods for parameterizing / classifying dietary variables for analysis is not clear. In particular, the authors talk about classifying quantiles of the baseline dietary variables for analysis, but present results for continuous models. The use of the 10-year follow-up data is also unclear, and it is unclear whether the small subset of participants included in the follow-up is representative and/or how diet reporting for this subset changed over the 10-year period.

Authors' Response:

On classifying dietary variables and use of the follow up data

To further clarify how we classified the dietary variables further (and in response to Reviewer #1s second comment in the Methods section) we have added more detail in our revised text. (Please see Authors' response to Reviewer #1's second comment for the Methods section.)

We have now also revised our text further to describe how we classified baseline dietary variables into categorical variables, and then used a continuous trend measure.

Statistical analysis, p. 4-5, Lines 154-166

For 61 foods and nutrients for which there was a quantitative measure of intake we calculated trends in risk of colorectal cancer per increment in grams, milligrams or micrograms per day using information collected in the Oxford WebQ (in women who had not developed colorectal cancer at the time of completing this). To do this all

baseline dietary intakes were first divided into categories, generally using quintiles; **for foods with non-continuous distributions we divided the dietary intakes into three to five categories using other appropriate cut-points.** We then derived repeat measures of intake within each baseline category by calculating the mean intakes for each food or nutrient category in women who had completed at least one 24-hour dietary assessment using the Oxford WebQ (based on the first completed Oxford WebQ where more than one had been completed), and assigning these mean intakes for all women in that category. **These re-measured mean intakes for each baseline category were then treated as a continuous variable in order to calculate log-linear trends in risk across categories of baseline intakes. The trend analyses used the baseline categories of intake defined in all women, but assigned the mean intakes for each of these baseline categories using the Oxford WebQ intakes measured during follow-up; this method does not alter the baseline categories, or the HRs for each category compared to the reference category, but provides better estimates of “usual” long-term dietary intakes during follow up and therefore better estimates of trends in HRs associated with defined increments in dietary exposures** (see Supplementary Methods for further description).

On representativeness

We agree with the reviewer that dietary intakes are expected to change over time. In response to Reviewer #1's second comment in the Methods section, we have now included revised text describing that within-person variability in dietary intake can be accounted for using a repeat sample in a sub-sample.

4. Analyses to address possible reverse causality are not clearly motivated. In particular, the authors conduct a sensitivity analysis excluding the first 5-years of follow-up, but it is unclear why they chose such a long interval of exclusion.

Authors' Response:

Our motivation for conducting a sensitivity analysis to assess the potential impact of reverse causation on our findings is the fact that early symptoms of bowel cancer could plausibly lead to changes in diet many years before a clinical diagnosis. While it is difficult to know precisely how long before diagnosis this might be expected to occur, early symptoms of disease could plausibly affect dietary choices as long as 5 years before diagnosis which is why we have chosen to exclude this period of follow-up in our sensitivity analyses. Few other studies have had sufficient power to exclude this duration of follow-up, to help ensure reliable examination of associations and identify dietary factors more likely to have a causal role in the risk of developing cancer. We have now described this in our revised text.

Methods, p. 6, Lines 221-225

Given that early symptoms of colorectal cancer could plausibly lead to changes in diet many years before diagnosis, we conducted sensitivity analyses restricted to women in self-reported excellent or good health at baseline, and to risk of colorectal cancer in the period five or more years after baseline, to assess the likely impact of potential reverse causation bias

Minor Comments:

5. It is not clear how the authors decided which of the 17 dietary factors in Table 3 to use as adjustment factors (other than milk and calcium). Perhaps fruit and whole grains were selected because are the only two “whole” food groups remaining? Clarity regarding this choice is needed.

Authors’ Response:

We additionally adjusted each of the 17 FDR dietary associations for calcium, dairy milk, fruit, and wholegrains because in addition to being the dietary factors that were most strongly related to risk, fruit and wholegrains were independent foods that were also highly correlated with other dietary factors (whereas for example red and processed meat intake was not highly correlated with the 16 other dietary factors so we did not expect that adjusting for this would affect the other associations). We have now clarified this in our revised text.

Methods, p. 5-6, Lines 204-205

We also investigated the degree to which each of the 17 dietary associations were independent of the four dietary factors **that were either the most strongly related to risk (calcium, dairy milk) or were foods that were substantially correlated with the other dietary factors (fruit, and wholegrains)**, by assessing the impact of further adjustment for each of these four factors individually.

6. Relative risk (RR) is used to refer to hazard ratio (HR) for most of the paper, but HR is used in a couple of places (Figure 1 and Supplementary Methods). It may be helpful to refer to HR first and then “hereafter referred to as relative risks”, or similar. Whichever terminology is used (RR or HR) should be consistent throughout the paper.

Authors’ Response:

We thank the reviewer for their comment and have now edited all HRs to RRs in the text, tables and figures and described this in the revised text.

Methods, p. 5, Lines 174-175

We used Cox proportional hazards regression models to estimate hazard ratios (hereafter referred to as relative risks) and 95% confidence intervals for associations between each of the 97 dietary factors and colorectal cancer incidence separately.

7. Line 186: Citation for residual method eventually goes back to Willett and Stampfer 1986 <https://academic.oup.com/aje/article/124/1/17/150019> (it seems like the paper cited here just used the method). Relatedly, line 189: why use quintiles and not just the residuals themselves (noted food intake values are divided into quintiles for some analyses, but the main findings in Table 3 are reported by trend increment)?

Authors’ Response:

We thank the reviewer for the reference. We now cite the Willett and Stampfer 1986 reference [19] in our revised text. With regards to the residuals, we included them in

our analyses as quintiles to avoid the need to make any assumptions about linearity and to minimise the potential influence of extreme values.

8. Line 337-338: “based on the low p-value for the association...” The low p-value could be due to large sample size, not necessarily reflective of the importance of the finding. In study of such large size, the authors should be cautious to interpret p-values outside the context of the magnitude of association.

Authors’ Response:

We agree with the reviewer that p value relates to sample size. However, the magnitude of the p value for the association of calcium with risk of colorectal cancer was much larger ($p < 0.000001$) than the association for dairy milk ($p < 0.00001$) or the other dairy-related foods and nutrients ($p < 0.0001$), which we think is important to note when trying to disentangle these highly correlated dietary factors.

9. Line 343: Having “The probable protective...” start a new paragraph may improve readability.

Authors’ Response:

Thank you we have started a new paragraph in the revised text.

10. Figure 1: what are the units for the foods/nutrients in the DWAS? As shown in Table 3?

Authors’ Response:

We agree that this could be clearer. We have now further described where to find the units in the revised text and added the units for the 17 FDR dietary factors in Figure 1.

Methods, p. 5, Line 168

Trend increments were selected based on the observed differences in WebQ derived intake between the lowest and highest baseline category (See increments for all in Supplementary Table 2).

Reviewer #3

References

1. Liu, B., H. Young, F.L. Crowe, V.S. Benson, E.A. Spencer, T.J. Key, P.N. Appleby, and V. Beral, *Development and evaluation of the Oxford WebQ, a low-cost, web-based method for assessment of previous 24 h dietary intakes in large-scale prospective studies*. Public Health Nutr, 2011. **14**(11): p. 1998-2005.
2. Swerdlow, D.I., K.B. Kuchenbaecker, S. Shah, R. Sofat, M.V. Holmes, J. White, J.S. Mindell, et al., *Selecting instruments for Mendelian randomization in the wake of genome-wide association studies*. Int J Epidemiol, 2016. **45**(5): p. 1600-1616.
3. MacMahon, S., R. Peto, J. Cutler, R. Collins, P. Sorlie, J. Neaton, R. Abbott, et al., *Blood pressure, stroke, and coronary heart disease. Part 1, Prolonged differences in blood pressure: prospective observational studies corrected for the regression dilution bias*. Lancet, 1990. **335**(8692): p. 765-74.
4. Key, T.J., A. Balkwill, K.E. Bradbury, G.K. Reeves, A.S. Kuan, R.F. Simpson, J. Green, and V. Beral, *Foods, macronutrients and breast cancer risk in postmenopausal women: a large UK cohort*. Int J Epidemiol, 2019. **48**(2): p. 489-500.

Response to reviewers

Re: Diet-wide analyses for risk of colorectal cancer: prospective study of 12,250 incident cases among 543,000 women in the UK

We thank the first reviewer for their additional comments, which we agree improve the manuscript. We have addressed Reviewer #1's comments point-by-point below. We present reviewer's comments below, followed by our responses (indented). Revised manuscript text appears as red text.

Reviewer reports	2
Reviewer #1	2

Reviewer reports

Reviewer #1

Thank you for the opportunity to review this revised manuscript. The authors have addressed most of the prior comments. However, a major concern remains regarding the methods used to estimate food and nutrient intake – it is still not clear how the intake was estimated. Additionally, it is questionable whether the method used is valid for estimating “long-term” or “usual intake.”

Authors' Response:

We thank the reviewer for their additional comments. We agree that this could be clearer. We have now provided additional information in response to the questions below in our revised text.

1. Specifically, the authors stated in their response to my question #2 (Methods) that they “calculated mean intakes of the 97 dietary factors from one web-based 24-hour dietary questionnaire.” However, based on Supplementary Table 1, the total number of dietary factors reported is 61, and some of these food items differ from those presented in Supplementary Table 2. It is confusing how the 97 dietary factors could be calculated using the WebQ method, which only captured 61 foods and nutrients.

Authors' Response:

We thank the reviewer for picking this up. We examined intake of 97 dietary factors, but only calculated mean intakes using the 24-hour recall for the 62 foods and nutrients which had quantitative information at baseline (i.e. those with more information than high versus low intake). For the other 35 binary dietary factors, we did not assign the 24-hour recall means. We have now clarified and corrected this in our revised text.

Methods, p. 4, Lines 123-126

In total, we included 97 dietary factors in our diet-wide analysis. Selection of foods and nutrients depended on their availability in both the dietary survey and the Oxford WebQ. **Of the 97 selected foods and nutrients, 62 were measured quantitatively and 35 were measured as binary intakes.** To enable calibrated estimates of intake to be made for all women **for the 62 foods and nutrients that were quantitatively assessed,** we first calculated the mean Oxford WebQ intake for each of the 62 dietary factors within each category (**e.g. quintiles**) of baseline intake for the sub-sample of women who completed the Oxford WebQ and then assigned these mean values to each baseline category for all women (see statistical analysis section for further details).

Statistical analysis, p. 4-5, Lines 148-173

For 62 foods and nutrients for which there was a quantitative measure of intake we calculated trends in risk of colorectal cancer per increment in grams, milligrams or micrograms per day using information collected in the Oxford WebQ (in women who had not developed colorectal cancer at the time of completing this). To do this, dietary intakes of these 62 foods and nutrients were first divided into categories, generally using quintiles;

.....The remaining 35 foods (which included fruit and vegetable subtypes, ice cream, legumes, and soy milk) were divided into two categories of baseline intake ('weekly' vs 'less than weekly'). We checked the consistency of high versus low intakes between the baseline and re-measured intakes for these foods, and calculated risk of colorectal cancer for high versus low intakes using the baseline intakes.

2. Lines 122-123 state: "Selection of foods and nutrients depended on their availability in both the dietary survey and the Oxford WebQ." Does this mean that only the overlapping food items or nutrients were included in the analysis? If so, it would be very helpful if the authors could show (perhaps using a Venn diagram) the number of overlapping food items measured using the two different dietary assessment tools.

Authors' Response:

That's correct, we only included foods and nutrients that were available in both dietary surveys. However, since both surveys were comprehensive, we ended up including all major foods groups and nutrients and as a result a very small number of foods were not included (e.g. aubergine) which are not likely to contribute meaningfully to dietary intakes or worth highlighting.

3. Lines 126-130 stated: "To enable calibrated estimates of intake to be made for all women, we first calculated the mean Oxford WebQ intake for each of the 97 dietary factors within each category of baseline intake for the sub-sample of women who completed the Oxford WebQ and then assigned these mean values to each baseline category for all women (see statistical analysis section for further details)." Would it be clearer to add in parentheses what the "category" refers to since "category" can also mean a category of food subgroups? For example, "...within each category (e.g., quintiles)...?"

Authors' Response:

We agree this would be clearer and have now added the example in our revised text.

Methods, p. 4, Lines 125-129

To enable calibrated estimates of intake to be made for all women for the 62 foods and nutrients that were quantitatively assessed, we first calculated the mean Oxford WebQ intake for each of the 62 dietary factors within each category (e.g. quintiles) of baseline intake for the sub-sample of women who completed the Oxford WebQ and then

assigned these mean values to each baseline category for all women (see statistical analysis section for further details)

4. Lines 149-151 stated: "...we divided the dietary intakes into three to five categories using other appropriate cut-points." Are there any justifications for choosing these cut points?

Authors' Response:

We based the cut points on the distribution of the data. We have now clarified this in our revised text.

Statistical analysis p. 4, Lines 153-154

To do this dietary intake of these 62 foods and nutrients were first divided into categories, generally using quintiles; for foods with non-continuous distributions, we divided the dietary intakes into three to five categories using other appropriate cut-points **to create approximately equal-sized groups based on the distribution of the data.**

5. The authors mentioned that they used "a statistical approach to estimate 'usual intake' during follow-up." Are there any references to support the validation of this approach to estimate "long-term" or "usual intake"? If not, I suggest the authors tone it down. It would be more appropriate to maintain transparency by clearly describing the methods used to obtain the estimations. Please also revise the abstract accordingly.

Authors' Response:

We agree that it would be difficult to ensure usual intake. However, we incorporated a repeat measure of dietary intake measured on average 10 years after the baseline, and therefore these intakes do reflect long-term dietary intakes. We have now softened our text (both in the abstract and the methods).

Abstract p. 2, Lines 40-41

Long-term intake within each baseline category was then estimated using information from a web-based 24-hour dietary assessment (conducted 10 years later, on average) which was available in a large subset of women, **to reduce the impact of measurement error and allow for changes in intake over time.**

Statistical analysis p.5, Line 163

The trend analyses used the baseline categories of intake defined in all women, but assigned the mean intakes for each of these baseline categories using the Oxford WebQ intakes measured during follow-up; this method does not alter the baseline categories, or the HRs for each category compared to the reference category, but provides better estimates of **"usual"** long-term dietary intakes during follow up and therefore better estimates of trends in HRs associated with defined increments in dietary exposures (see Supplementary Methods for further description).

Discussion p.5, Lines 445-446

In addition, we used a web-based 24-hour dietary questionnaire (the Oxford WebQ), validated against recovery biomarkers [64], to re-measure diet about 10 years later to estimate ~~usual~~ long-term diet and test for trends across baseline categories of intake; ~~thus reducing the impact of measurement error and changes in diet over time~~ [15].

6. Regarding Table 3, please specify how the exposure was modeled. Was it modeled as a category variable based on the baseline categories? Or was it modeled as a continuous variable using the increment defined in column 2 “trend increment”? Please also make sure this is clearly described in the "statistical method" section.

Authors' Response:

We modelled the exposures in Table 3 as log-linear trends in risk across categories of baseline intakes for all 17 foods and nutrients using the increments listed for each dietary factor. We have now clarified this in the statistical analysis section and in the Table and Figure 1 footnotes.

Statistical analysis p.5-6, Lines 201-202

To investigate the potential role of confounding by lifestyle factors in the associations between these 17 dietary factors and colorectal cancer, ~~modelled as log-linear trends in risk across categories of baseline intakes~~, we calculated the change in the log relative risk associated with each of the 17 dietary factors after differing levels of adjustment for potential confounders.

Figure 1 and Supplementary Table 1 footnote

~~For each of the 62 quantitatively measured dietary factors, we created a continuous variable using the re-measured mean intakes for each baseline category. Log-linear trends in risk across categories of baseline intakes were then calculated using the listed increments.~~

Table 3 footnote

~~For each dietary factor, we created a continuous variable using the re-measured mean intakes for each baseline category. Log-linear trends in risk across categories of baseline intakes were then calculated using the listed increments.~~